# JunB regulates homeostasis and suppressive functions of effector regulatory T cells

Shin-ichi Koizumi[1], Daiki Sasaki[1], Tsung-Han Hsieh [ID] [1], Naoyuki Taira[1], Nana Arakaki[2], Shinichi Yamasaki[2], Ke Wang[1], Shukla Sarkar[1], Hiroki Shirahata[1], Mio Miyagi[1] & Hiroki Ishikawa[1]

Foxp3-expressing $CD4^+$ regulatory T (Treg) cells need to differentiate into effector Treg (eTreg) cells to maintain immune homeostasis. T-cell receptor (TCR)-dependent induction of the transcription factor IRF4 is essential for eTreg differentiation, but how IRF4 activity is regulated in Treg cells is still unclear. Here we show that the AP-1 transcription factor, JunB, is expressed in eTreg cells and promotes an IRF4-dependent transcription program. Mice lacking JunB in Treg cells develop multi-organ autoimmunity, concomitant with aberrant activation of T helper cells. JunB promotes expression of Treg effector molecules, such as ICOS and CTLA4, in BATF-dependent and BATF-independent manners, and is also required for homeostasis and suppressive functions of eTreg. Mechanistically, JunB facilitates the accumulation of IRF4 at a subset of IRF4 target sites, including those located near *Icos* and *Ctla4*. Thus, JunB is a critical regulator of IRF4-dependent Treg effector programs, highlighting important functions for AP-1 in Treg-mediated immune homeostasis.

[1] Immune Signal Unit, Okinawa Institute of Science and Technology Graduate University, 1919-1 Tancha, Onna-son, Okinawa 904-0495, Japan. [2] DNA Sequencing Section, Okinawa Institute of Science and Technology Graduate University, 1919-1 Tancha, Onna-son, Okinawa 904-0495, Japan. These authors contributed equally: Shin-ichi Koizumi, Daiki Sasaki. Correspondence and requests for materials should be addressed to H.I. (email: hiroki.ishikawa@oist.jp)

Regulatory T (Treg) cells can suppress a variety of immune responses and contribute to immune homeostasis[1,2]. They require the lineage-specifying transcription factor, forkhead box P3 (Foxp3), for development, maintenance of cell identity, and suppressive functions[3–5]. Thymic Treg (tTreg) cells develop in the thymus and then circulate through blood and lymphoid tissues as central Treg (cTreg) cells, displaying a CD62L[high (hi)] phenotype[6]. While circulating in the periphery, some cTreg cells differentiate into effector Treg (eTreg) cells, exhibiting an activated cell phenotype (CD62L[low (lo)]), depending on T-cell receptor (TCR) stimuli[7]. cTreg and eTreg cells express various genes differentially and perform non-redundant functions[6,8–13]. For example, cTreg cells express lymph node-homing receptors CD62L and CC chemokine receptor 7 (CCR7), whereas eTreg cells preferentially express other chemokine receptors and accumulate in non-lymphoid tissues[14,15]. In addition, eTreg cells express higher levels of Treg effector molecules, such as inducible T-cell costimulator (ICOS) and cytotoxic T-cell-associated antigen 4 (CTLA4), than do cTreg cells, which are likely important for eTreg-suppressive functions and homeostasis[7,14,16–18]. Furthermore, eTreg cells can express transcription factors, T-box transcription factor (T-bet), GATA-binding protein 3 (GATA3), and retinoic acid receptor-related orphan receptor (ROR)γt, which are associated with roles unique to eTreg cells, to regulate functions of distinct T helper cells[19–24].

To activate the effector program, Treg cells require TCR-dependent transcriptional regulatory mechanisms that CD4⁺Foxp3⁻ conventional T (Tconv) cells also use for differentiation[12,25–27]. For example, the transcription factor, interferon regulatory factor 4 (IRF4), which promotes differentiation of several types of T helper cells and cytotoxic CD8⁺ T cells[28–32], is induced in Treg cells upon TCR stimulation and participates in eTreg differentiation[7,33]. Treg-specific Irf4-deficient mice develop multi-organ autoimmunity, exhibiting profound Th2 responses[34]. Furthermore, basic leucine zipper transcription factor ATF-like (BATF), one of the AP-1 family of transcription factors, which is also essential for differentiation of Th17 cells and cytotoxic CD8⁺ T cells[35,36], is highly expressed in eTreg cells, and Batf-deficient Treg cells cannot suppress T-cell-mediated colitis[24]. BATF and IRF4 bind to AP-1-IRF-composite element (AICE) motifs[37–40], but how BATF and other AP-1 transcription factors regulate IRF4 activity in the eTreg transcriptional program is not fully understood.

JunB is another member of the AP-1 family of transcription factors[37]. AP-1 factors contain basic leucine zipper (bZIP) domains, by which they form homodimers or heterodimers with other AP-1 or bZIP-containing transcription factors[41]. JunB can dimerize with BATF in a variety of cells, including CD8⁺ T cells and Th17 cells[37]. We and other groups recently reported that JunB regulates the BATF- and IRF4-dependent Th17 cell transcriptional program that is critical for pathogenic functions of Th17 cells[42–45]. JunB promotes DNA-binding of BATF and IRF4 at IRF4 target sites, including those associated with Th17-related genes, Rorc and Il23 receptor (Il23r), during Th17 differentiation[42,43]. However, functions of JunB in Treg cells remain unclear.

Here, we show that JunB is expressed in eTreg cells, and is required for eTreg-mediated immune homeostasis. Treg-specific deletion of JunB induces severe inflammation in lung and colon. JunB facilitates expression of ICOS and CTLA4 in BATF-dependent and -independent fashions, and is pivotal for homeostasis and suppressive functions of eTreg cells. Mechanistically, JunB is required for DNA-binding of IRF4 at IRF4 target sites associated with Icos and Ctla4. Thus, we show crucial functions for JunB in the IRF4-dependent eTreg transcriptional program.

## Results

### JunB expression in Treg cells

To understand JunB functions in Treg cells, we first analyzed expression of JunB in murine Treg cells by flow cytometry. In spleen, a subset of Foxp3⁺ Treg cells, but not Foxp3⁻ Tconv cells, expressed substantial levels of JunB (Fig. 1a and Supplementary Fig. 1a). Moreover, in lung, Treg cells, as well as Tconv cells, uniformly expressed high levels of JunB, and the expression level of JunB in Treg cells was significantly higher than in Tconv cells (Fig. 1b). We next assessed whether there is a correlation between JunB expression and distinct Treg subpopulations. cTreg cells circulate through secondary lymphoid organs, whereas eTreg cells preferentially accumulate in peripheral tissues[6,14,24]. We analyzed JunB expression in CD62L[hi]CD44[lo] cTreg cells and CD62L[lo] eTreg cells (Supplementary Fig. 1b). Consistent with enrichment of JunB-expressing Treg cells in the lung, we found that JunB was expressed in eTreg cells, but not in cTreg cells in spleen (Fig. 1c). eTreg cells heterogeneously express surface molecules such as ICOS, T-cell immunoreceptor with Ig and ITIM domains (TIGIT), and killer cell lectin-like receptor 1 (KLRG1)[11,18,24,46–49]. We noted that JunB expression was correlated with expression of ICOS, TIGIT, and KLRG1 (Fig. 1d and Supplementary Fig. 1c, d). As reported previously, IRF4 and BATF[24,33,34], which are required for eTreg differentiation, were also expressed at elevated levels in ICOS-expressing eTreg cells (Supplementary Fig. 1e, f). JunB expression can be regulated transcriptionally and post-transcriptionally in various signaling pathways[50,51]. To determine whether induction of JunB is transcriptionally regulated in eTreg cells, we sorted cTreg and eTreg cells by fluorescence-activated cell sorting (FACS) and analyzed expression of mRNA for Junb, as well as Batf and Irf4, by reverse transcriptase quantitative polymerase chain reaction (RT-qPCR) analysis. Although mRNA expression of Batf and Irf4 was upregulated in eTreg cells, there was no difference of Junb mRNA expression between cTreg and eTreg cells (Fig. 1e), suggesting that, unlike BATF and IRF4, JunB expression is regulated post-transcriptionally in eTreg cells. These data indicate that JunB is expressed in a subset of eTreg cells.

To investigate how JunB expression is regulated in Treg cells, we examined expression of JunB, as well as of BATF and IRF4, in TCR-stimulated Treg cells, because TCR signaling is necessary for differentiation of eTreg cells[7,52]. We isolated CD4⁺CD25⁺ Treg cells from spleens and confirmed that > 95% of the cells expressed Foxp3 (Supplementary Fig. 1g). We activated Treg cells with anti-CD3 and anti-CD28 antibodies in the presence of interleukin (IL)−2. Flow cytometry analysis showed that expression of JunB and BATF was induced by both anti-CD28 antibody and IL-2 stimulation in an additive manner, compared with expression levels in Treg cells stimulated with anti-CD3 antibody alone (Fig. 1f). On the other hand, IRF4 expression was markedly induced by stimulation with anti-CD3 antibody alone, and it was further enhanced by either anti-CD28 antibody or IL-2 stimulation (Fig. 1f). However, the additive effect of anti-CD28 antibody and IL-2 stimulation was not observed in IRF4 expression (Fig. 1f). In summary, these results suggest that dynamic expression of JunB in TCR-stimulated Treg cells might regulate generation and/or function of eTreg cells.

### Treg-specific deletion of JunB induces autoimmunity

To investigate physiological functions of JunB in Treg cells, we crossed mice harboring loxp-flanked Junb alleles (Junb[fl/fl]) with mice harboring Foxp3 promoter-driven cre recombinase (Foxp3[Cre]). This generated Treg-specific, Junb-deficient (Foxp3[Cre]Junb[fl/fl]) mice (Supplementary Fig. 2a). Flow cytometry analysis confirmed efficient deletion of JunB in CD4⁺Foxp3⁻ Treg cells in Foxp3[Cre]Junb[fl/fl] mice (Supplementary Fig. 2a). In

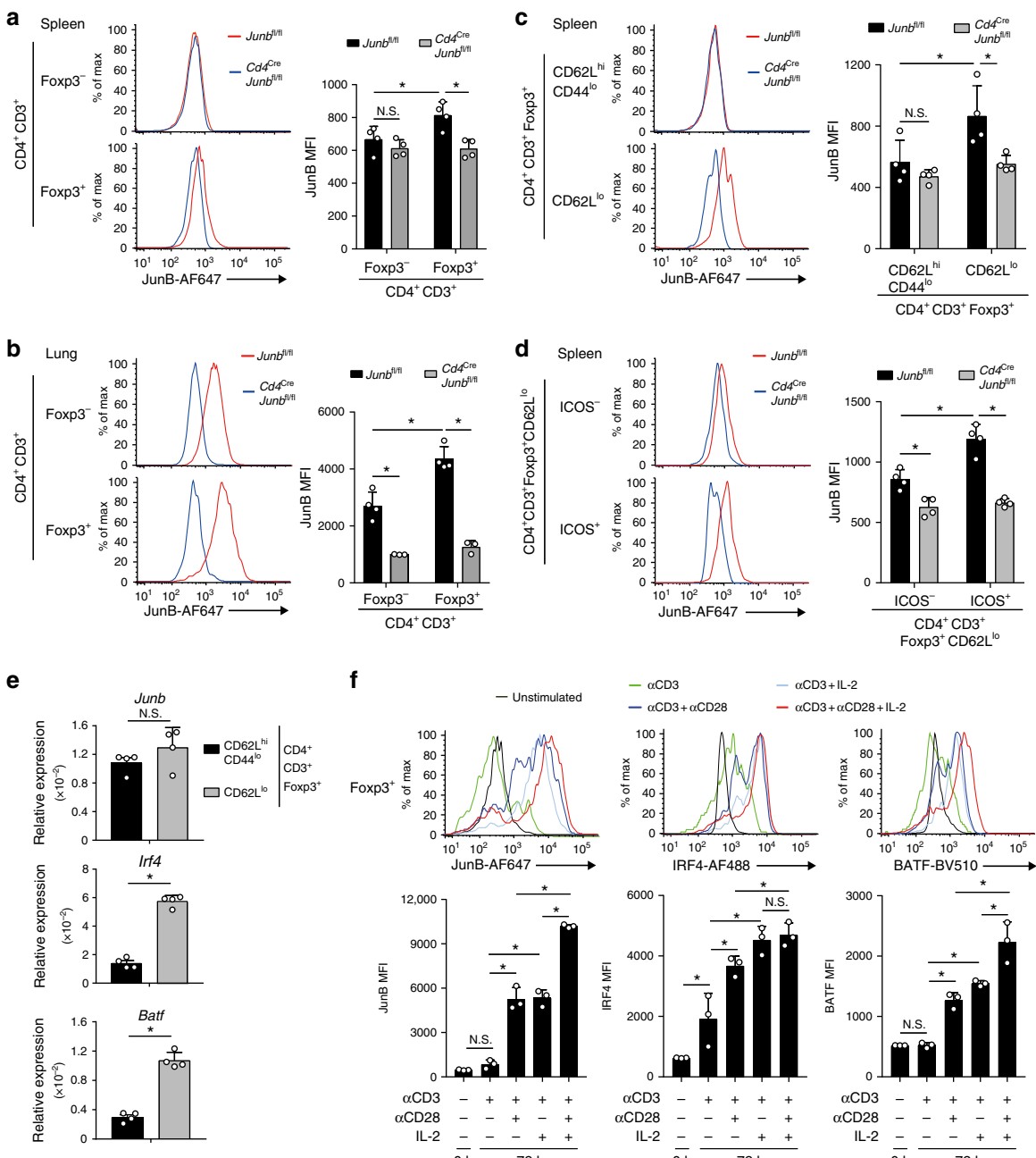

**Fig. 1** Expression of JunB is upregulated in eTreg cells. **a–d** Flow cytometry analysis of JunB in Foxp3$^+$ (Treg) or Foxp3$^-$ (Tconv) cells isolated from spleen **a** and lung **b**, Treg cells bearing CD62L$^{hi}$CD44$^{lo}$ phenotypes (cTreg) or CD62L$^{lo}$ phenotypes (eTreg) **c**, and ICOS$^+$ or ICOS$^-$ eTreg cells **d** isolated from spleen of wild-type C57BL/6 mice (7–10-week-old). $Cd4^{cre}Junb^{fl/fl}$ mice were used as negative controls. **e** CD62L$^{hi}$CD44$^{lo}$ cTreg and CD62L$^{lo}$ eTreg cells were sorted by FACS, and $Junb$ mRNA expression was analyzed by qRT-PCR. **a–e** Error bars indicate s.d. ($n = 4$). *$P < 0.05$; N.S., not significant (unpaired two-tailed Student's $t$ test). MFI, mean fluorescence intensity. **f** JunB expression was analyzed by flow cytometry in CD4$^+$CD25$^+$ Treg cells activated with indicated stimuli for 72 h. Error bars indicate s.d. ($n = 3$). *$P < 0.05$; N.S., not significant (unpaired two-tailed Student's $t$ test). Data represent two independent experiments

contrast, substantial numbers of CD4$^+$Foxp3$^-$ Tconv cells and CD8$^+$ T cells expressed JunB in the lung and colon of $Foxp3^{Cre}Junb^{fl/fl}$ mice (Supplementary Fig. 2a), albeit at a lower number compared with control mice, probably owing to leaky expression of Cre in these cells[53]. $Foxp3^{Cre}Junb^{fl/fl}$ mice were born at normal mendelian ratios, but they were markedly smaller, and by 4 weeks of age their weights were already significantly lower than those of control mice (Fig. 2a and Supplementary Fig. 2b). About 60% of $Foxp3^{Cre}Junb^{fl/fl}$ mice died within

6 months (Fig. 2b). We observed significant increases in size and cellularity of cervical lymph nodes, but not other lymph nodes and spleens in $Foxp3^{Cre}Junb^{fl/fl}$ mice, compared with $Foxp3^{Cre}-Junb^{+/+}$ mice (Fig. 2c and Supplementary Fig. 2c). In histopathological analysis, $Foxp3^{Cre}Junb^{fl/fl}$ mice exhibited higher inflammation scores in lung and colon (but not in liver or skin), than did control mice (Fig. 2d).

In $Foxp3^{Cre}Junb^{fl/fl}$ mice, although increased cellularity was restricted to cervical lymph nodes (Fig. 2c), frequencies of

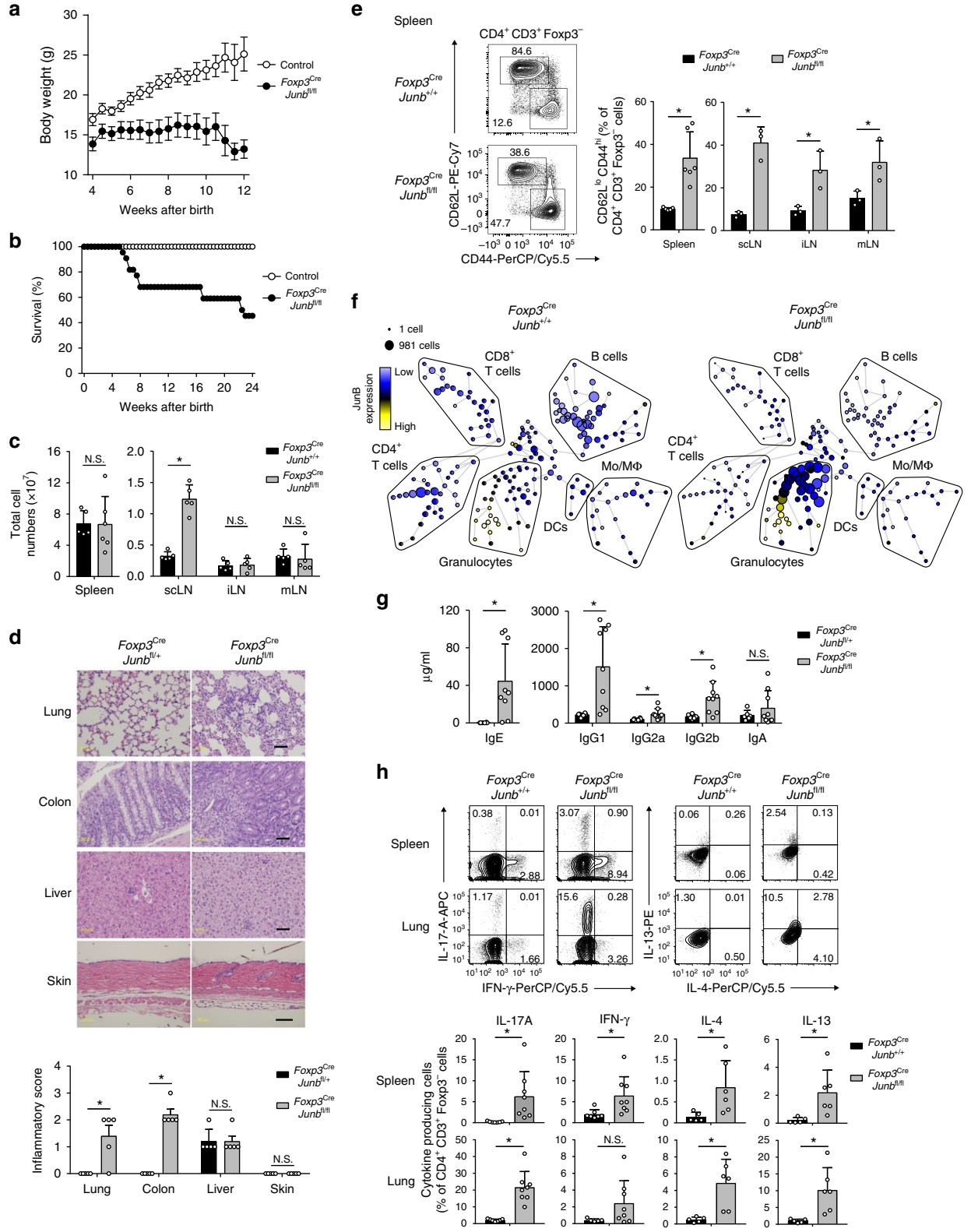

CD4[+]Foxp3[−] Tconv cells exhibiting the activated cell phenotype (CD62L[lo]CD44[hi]) were markedly more abundant than in *Foxp3*[Cre]*Junb*[+/+] mice in all secondary lymphoid tissues analyzed and in lung (Fig. 2e, Supplementary Fig. 2d). Total B cells (B220[+]CD19[+]), germinal center B cells (B220[+]CD19[+]GL-7[+]Fas[+]), and plasma cells (CD138[+]B220[−]) were also significantly increased in the cervical lymph nodes, but not in the spleen or Peyer's patches, in

*Foxp3*[Cre]*Junb*[fl/fl] mice, compared with *Foxp3*[Cre]*Junb*[+/+] mice (Supplementary Fig. 3a–c). Furthermore, mass cytometry analysis revealed increased frequencies of granulocytes with unaltered JunB expression levels in these cells in spleens of *Foxp3*[Cre]*Junb*[fl/fl] mice (Fig. 2f), indicating that JunB in Treg cells is important for maintenance of leukocyte homeostasis. We also analyzed levels of serum immunoglobulins by enzyme-linked immunosorbent assay

**Fig. 2** Mice deficient for JunB in Treg cells develop autoimmune disease. **a** Body weight changes of male $Foxp3^{Cre}Junb^{fl/fl}$ mice and control mice. Error bars indicate s.e.m. ($n = 11$ for controls, $n = 10$ for $Foxp3^{Cre}Junb^{fl/fl}$ mice). **b** Survival of male $Foxp3^{Cre}Junb^{fl/fl}$ mice and control mice. Error bars indicate s.d. ($n = 11$ for controls, $n = 22$ for $Foxp3^{Cre}Junb^{fl/fl}$ mice). **c** Total cell numbers in spleens, superficial cervical lymph nodes (scLN), inguinal LN (iLN), and mesenteric LN (mLN) of male $Foxp3^{Cre}Junb^{fl/fl}$ mice and control mice (8–12-week-old). Error bars indicate s.d. ($n = 5$). *$P < 0.05$; N.S., not significant (unpaired two-tailed Student's $t$ test). **d** Hematoxylin and eosin staining of lung, colon, liver, and skin from 12-week-old male $Foxp3^{Cre}Junb^{fl/fl}$ mice and control mice. Scale bars, 50 μm (lung, colon, and liver) or 100 μm (skin). Bar graph shows histopathological inflammation scores. Error bars indicate s.d. ($n = 5$). *$P < 0.05$; N.S., not significant (unpaired two-tailed Student's $t$ test). **e** Flow cytometry analysis of CD62L and CD44 in CD4+Foxp3− Tconv cells isolated from various tissues of male $Foxp3^{Cre}Junb^{fl/fl}$ mice and $Foxp3^{Cre}Junb^{+/+}$ mice (8–12-week-old). Representative flow cytometry profiles show CD4+Foxp3− Tconv cells isolated from the spleen. The graph shows percentages of CD62L$^{lo}$CD44$^{hi}$ activated cells among CD4+Foxp3− cells. Error bars indicate s.d. ($n = 5$ for spleens, $n = 3$ for lymph nodes). *$P < 0.05$; N.S., not significant (unpaired two-tailed Student's $t$ test). **f** Mass cytometry analysis of leukocytes isolated from spleens of $Foxp3^{Cre}Junb^{fl/fl}$ mice and $Foxp3^{Cre}Junb^{+/+}$ mice (8–12-week-old). Mo: monocytes, Mφ: macrophages, DC: dendritic cells. Size of each node represents cell numbers. Expression levels of JunB is color-coded. **g** ELISA analysis of immunoglobulin isotypes in sera of 12-week-old male $Foxp3^{Cre}Junb^{fl/fl}$ mice and $Foxp3^{Cre}Junb^{+/+}$ mice. Error bars indicate s.d. ($n = 6$ for $Foxp3^{Cre}Junb^{+/+}$ mice, $n = 9$ for $Foxp3^{Cre}Junb^{fl/fl}$ mice). *$P < 0.05$; N.S., not significant (unpaired two-tailed Student's $t$ test). **h** Flow cytometry analysis of intracellular IL-17A, IFN-γ, IL-4, and IL-13 in CD4+Foxp3− cells isolated from spleens of 8–12-week-old male $Foxp3^{Cre}Junb^{fl/fl}$ mice and $Foxp3^{Cre}Junb^{+/+}$ mice. Error bars indicate s.d. ($n = 5$). *$P < 0.05$ (unpaired two-tailed Student's $t$ test). Data represent two independent experiments

(ELISA) and found that $Foxp3^{Cre}Junb^{fl/fl}$ mice produced significantly elevated levels of serum IgG1, IgG2a, IgG2b, and IgE, but not IgA (Fig. 2g).

To explore whether JunB is important for Treg cells to suppress functions of specific T helper cells, we analyzed expression of signature cytokines produced by distinct T helper cells (interferon (IFN)-γ for Th1 cells, IL-4 and IL-13 for Th2 cells, and IL-17A for Th17 cells). In spleen and lung, $Foxp3^{Cre}Junb^{fl/fl}$ mice exhibited more abundant cells expressing IFN-γ, IL-4, IL-13, and IL-17A than did $Junb^{fl/fl}$ mice (Fig. 2h). We also found that CD8+ T cells with an activated phenotype (CD62L$^{lo}$CD44$^{hi}$) or expression of inflammatory cytokines (IFN-γ and IL-17A) were significantly increased in spleen and lung of $Foxp3^{Cre}Junb^{fl/fl}$ mice, compared with control mice (Supplementary Fig. 3d, e). Thus, loss of JunB in Treg cells induces multi-organ autoimmune pathology accompanied by activation of T helper cells, CD8+ T cells and B cells, dysregulated leukocyte homeostasis in lymphoid tissues, and enhanced production of immunoglobulins.

**JunB regulates accumulation and function of Treg cells.** We next investigated whether autoimmune phenotypes of $Foxp3^{Cre}Junb^{fl/fl}$ mice are due to loss of Treg cells. First, we analyzed abundances of Treg cells in various tissues. In colon of $Foxp3^{Cre}Junb^{fl/fl}$ mice, CD4+Foxp3+ Treg cells were significantly decreased in both absolute numbers and proportions among T cells (Fig. 3a), suggesting that a decrease of Treg cells might partly explain the more severe inflammation observed in the colon of $Foxp3^{Cre}Junb^{fl/fl}$ mice. In contrast, abundance of CD4+Foxp3+ Treg cells was normal in lung, liver, and skin, and only slightly reduced in spleen in $Foxp3^{Cre}Junb^{fl/fl}$ mice (Fig. 3a and Supplementary Fig. 4a), suggesting that inflammation induced in lung and spleen of the mice might not be due to loss of Treg cells.

To further explore why Treg-mediated self-tolerance cannot be maintained in $Foxp3^{Cre}Junb^{fl/fl}$ mice, we next analyzed abundance of eTreg cells, as JunB was expressed in this population. $Foxp3^{Cre}Junb^{fl/fl}$ mice had normal or even increased numbers of CD62L$^{lo}$ eTreg cells in the spleen and lung, but not in the colon, compared with $Foxp3^{Cre}Junb^{+/+}$ mice (Fig. 3b and supplementary Fig 4b, c). However, the mean fluorescence intensity (MFI) of CD44 was reduced in $Junb$-deficient eTreg cells (Fig. 3b, c). We also analyzed expression of Treg signature molecules and found that there was no reduction in expression of CTLA4 and CD25 in eTreg and cTreg cells of $Foxp3^{Cre}Junb^{fl/fl}$ mice (Fig. 3d). GITR (Glucocorticoid-Induced TNF receptor family-related protein) was even upregulated in both eTreg and cTreg cells of $Foxp3^{Cre}Junb^{fl/fl}$ mice, probably owing to effects of inflammation

(Fig. 3d). Moreover, we analyzed expression of eTreg-related molecules and found that in $Junb$-deficient eTreg cells, ICOS expression was severely diminished, but expression of TIGIT, KLRG1, and ST2 (IL-33 receptor) was not affected (Fig. 3e). These data indicate that JunB is not required for generation of CD62L$^{lo}$ eTreg cells, but it promotes expression of CD44 and ICOS in eTreg cells in diseased $Foxp3^{Cre}Junb^{fl/fl}$ mice.

A subset of eTreg cells expresses transcription factors that have lineage-defining functions in T helper cells, such as T-bet, GATA3, and RORγt[19–24]. These transcription factors likely regulate migration and suppressive functions of eTreg cells in a context-dependent manner. We analyzed expression of these transcription factors in Treg cells of $Foxp3^{Cre}Junb^{fl/fl}$ mice. There was a significant reduction of GATA3 and T-bet, but not RORγt, in Treg cells in spleens of $Foxp3^{Cre}Junb^{fl/fl}$ mice compared with $Foxp3^{Cre}Junb^{+/+}$ mice (Supplementary Fig. 4d).

To directly examine the functional importance of JunB in Treg cell immune-suppressive activity, we performed an in vitro suppression assay. $Junb$-deficient Treg cells isolated from $Cd4^{Cre}Junb^{fl/fl}$ mice or $Junb$-sufficient Treg cells isolated from $Junb^{fl/fl}$ mice were mixed with activated Tconv cells. Cell trace violet (CTV) staining analysis showed that suppressive activity of $Junb$-deficient Treg cells was significantly impaired (Fig. 3f). Moreover, pre-activation of Treg cells with anti-CD3 antibody enhanced suppression activity of $Junb$-sufficient Treg cells, but not $Junb$-deficient Treg cells (Fig. 3f). Notably, upon stimulation with anti-CD3 antibody, there was a significant increase of annexin-V+ cells in $Junb$-deficient Treg cells, compared with $Junb$-sufficient Treg cells (Supplementary Fig. 5a). These data suggest that JunB is required for survival of Treg cells upon prolonged and/or strong TCR stimulation, which enhances suppressive activity of Treg cells.

We also performed an in vivo suppression assay by co-transferring $Junb$-deficient Treg cells (CD4+CD25+) isolated from $Cd4^{Cre}Junb^{fl/fl}$ or $Foxp3^{Cre}Junb^{fl/fl}$ mice with wild-type naive CD4+ T cells into $Rag1$-deficient mice. Unlike $Junb$-sufficient Treg cells, $Junb$-deficient Treg cells could not suppress weight loss and intestinal pathology in these mice (Fig. 3g and supplementary Fig. 5b). Forty days after transfer, there was no difference in the frequency of $Junb$-deficient and -sufficient Treg cells in the spleen and lung (Supplementary Fig. 5c). However, in the mesenteric lymph nodes and colon, frequency of $Junb$-deficient Treg cells was significantly lower than that of $Junb$-sufficient Treg cells (Supplementary Fig. 5c). In summary, these data suggest that JunB supports tissue-specific accumulation and suppressive functions of Treg cells.

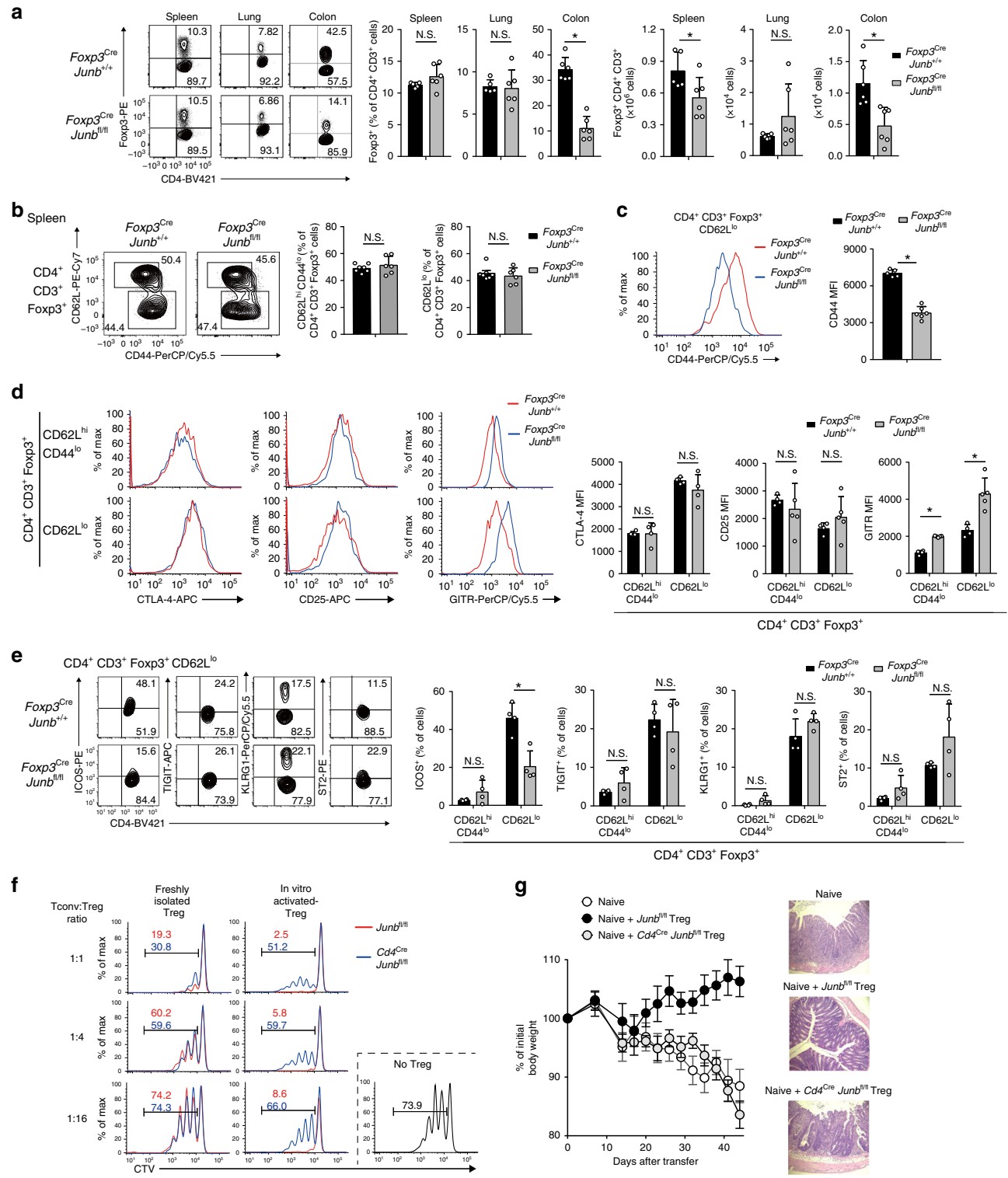

**JunB regulates expression of Treg effector genes**. In *Foxp3*^Cre*Junb*^fl/fl mice, *Junb*-deficient Treg cell phenotypes may be modulated by inflammation. To elucidate JunB function in Treg cells under homeostatic (non-inflammatory) conditions, we used *Cd4*^Cre*Junb*^fl/fl mice, which show no inflammatory signs, probably owing to impaired activation of T helper cells[42–44]. We first analyzed Treg cell abundance in various lymphoid and non-lymphoid tissues. Consistent with recent reports[43], CD4⁺Foxp3⁺ Treg cells were diminished in all tissues examined in

*Cd4*^Cre*Junb*^fl/fl mice, with the greatest reduction in colon (Fig. 4a). This implies that JunB is required for the accumulation of Treg cells under non-inflammatory conditions.

We then analyzed eTreg cell abundance in *Cd4*^Cre*Junb*^fl/fl mice. In spleens of *Cd4*^Cre*Junb*^fl/fl mice, the proportion of CD62L^lo eTreg cells among all Foxp3⁺ Treg cells was lower than in *Junb*^fl/fl mice (Fig. 4b). As in diseased *Foxp3*^Cre*Junb*^fl/fl mice, MFIs of CD44 expression in CD62L^lo eTreg cells of *Cd4*^Cre*Junb*^fl/fl mice were significantly lower than in *Junb*^fl/fl mice (Fig. 4c). This

**Fig. 3** JunB is required for accumulation and effector functions of Treg cells. **a** Flow cytometry analysis of Foxp3 in CD4$^+$ T cells in spleens, lungs, and colons of male $Foxp3^{Cre}Junb^{fl/fl}$ and $Foxp3^{Cre}Junb^{+/+}$ mice (8–12-week-old). Graphs show percentages and numbers of CD4$^+$Foxp3$^+$ Treg cells. Error bars indicate s.d. ($n = 5$ for $Foxp3^{Cre}Junb^{+/+}$ mice, $n = 6$ for $Foxp3^{Cre}Junb^{fl/fl}$ mice). *$P < 0.05$; N.S., not significant (unpaired two-tailed Student's $t$ test). **b**, **c** Flow cytometry analysis of CD44 and CD62L in CD4$^+$Foxp3$^+$ Treg cells isolated from spleens of male $Foxp3^{Cre}Junb^{fl/fl}$ and $Foxp3^{Cre}Junb^{+/+}$ mice (8–12-week-old). Graph shows percentages of CD62$^{hi}$CD44$^{lo}$ cTreg cells and CD62$^{lo}$ eTreg cells **b**, and MFIs of CD44 **c**. Error bars indicate s.d. ($n = 7$ for $Foxp3^{Cre}Junb^{+/+}$ mice, $n = 6$ for $Foxp3^{Cre}Junb^{fl/fl}$ mice). *$P < 0.05$; N.S., not significant (unpaired two-tailed Student's $t$ test). **d**, **e** Flow cytometry analysis of CTLA4, CD25, and GITR **d**, and ICOS, TIGIT, KLRG1, and ST2 **e** in CD62$^{hi}$CD44$^{lo}$ cTreg cells and CD62$^{lo}$ eTreg cells among CD4$^+$Foxp3$^+$ Treg cells isolated from spleens of male $Foxp3^{Cre}Junb^{fl/fl}$ and $Foxp3^{Cre}Junb^{+/+}$ mice (8–12-week-old). Representative flow cytometry profiles show CD62$^{hi}$CD44$^{lo}$ cTreg cells and CD62$^{lo}$ eTreg cells **d**, and CD62$^{lo}$ eTreg cells **e**. Graphs show MFIs of CTLA4, CD25, and GITR **d**, and percentages of cells expressing indicated molecules **e**. Error bars indicate s.d. ($n = 4$). *$P < 0.05$; N.S., not significant (unpaired two-tailed Student's $t$ test). **f** CD4$^+$CD25$^+$ Treg cells were isolated from $Cd4^{Cre}Junb^{fl/fl}$ and $Junb^{fl/fl}$ mice. Treg cells freshly isolated or activated with anti-CD3 and anti-CD28 antibodies in the presence of IL-2 for 3 days were used for in vitro suppression assay. CD4$^+$CD25$^-$ Tconv cells (CD45.1) labeled with cell trace violet (CTV) were cultured with different numbers of CD4$^+$CD25$^+$ Treg cells in the presence of anti-CD3-/anti-CD28-coated beads for 2 days, and CTV dilution was analyzed by flow cytometry. **g** In vivo suppression assay using CD4$^+$CD25$^+$ Treg cells isolated from $Cd4^{Cre}Junb^{fl/fl}$ and $Junb^{fl/fl}$ mice. In all, 6–8-week-old sex-matched $Rag1^{-/-}$ mice were injected with wild-type naive CD4$^+$ T cells and CD4$^+$CD25$^+$ Treg cells. The graph shows body weight changes. Colonic histopathology was analyzed on day 40 after injection. Error bars indicate s.e.m. ($n = 5$). Data represent two independent experiments

suggests that JunB is involved in accumulation of eTreg cells under homeostatic conditions.

We next assessed expression of signature molecules for Treg cells and eTreg cells in $Cd4^{Cre}Junb^{fl/fl}$ mice. Expression levels of CTLA4, but not of CD25 and GITR, in eTreg cells were significantly lower in $Cd4^{Cre}Junb^{fl/fl}$ mice than $Junb^{fl/fl}$ mice (Fig. 4d). In addition, there was a significant reduction of CD62L$^{lo}$ eTreg cells expressing ICOS, TIGIT, and KLRG1, but not ST2, in spleens of $Cd4^{Cre}Junb^{fl/fl}$ mice (Fig. 4e). Although we detected decreases of GATA3 and T-bet in eTreg cells of $Foxp3^{Cre}Junb^{fl/fl}$ mice, those molecules were normally expressed in eTreg cells of $Cd4^{Cre}Junb^{fl/fl}$ mice (Supplementary Fig. 6a), suggesting that JunB is involved in expression of GATA3 and T-bet only under inflammatory conditions. Assay with antibodies against neurophilin 1 (Nrp1) or Helios, which are markers for Treg cells generated in the thymus (tTreg), revealed that ICOS expression was significantly decreased in Treg populations, regardless of their expression of Nrp1 or Helios (Fig. 4f and Supplementary Fig. 6b), suggesting that JunB promotes ICOS expression in both tTreg cells and Treg cells induced in periphery (pTreg). We also analyzed expression of eTreg signature molecules in neonates, because Treg cells generated at that age perform unique and essential functions to maintain self-tolerance[54]. There was no difference in frequency of Treg cells in the thymus between 1-week-old $Cd4^{Cre}Junb^{fl/fl}$ and $Junb^{fl/fl}$ neonates (Supplementary Fig. 6c). However, expression of ICOS, TIGIT, and KLRG1 in splenic Treg cells was severely decreased in 1-week-old $Cd4^{Cre}Junb^{fl/fl}$ mice (Fig. 4g). Taken together, JunB is needed for expression of a subset of effector molecules in eTreg cells and for accumulation of eTreg cells under homeostatic conditions.

**JunB regulates eTreg homeostasis**. We next examined whether cell-intrinsic functions of JunB are required for generation and homeostasis of Treg cells. To this end, we transferred bone marrow (BM) cells from $Cd4^{Cre}Junb^{fl/fl}$ and $Junb^{fl/fl}$ mice (CD45.2$^+$), in combination with equal numbers of wild-type (WT) BM cells (CD45.1$^+$), into $Rag1^{-/-}$ mice, which generated WT:$Cd4^{Cre}Junb^{fl/fl}$ and WT:$Junb^{fl/fl}$ BM chimeric mice, respectively (Fig. 5a). These BM chimeric mice had comparable numbers of total Foxp3$^+$ Treg cells in all tissues examined (Supplementary Fig. 7). We then analyzed ratios of cells derived from $Cd4^{Cre}Junb^{fl/fl}$ and $Junb^{fl/fl}$ BM cells (CD45.2$^+$) relative to those derived from WT BM cells (CD45.2$^-$) among Treg populations. In the thymus, frequencies of Foxp3$^+$ Treg cells derived from $Cd4^{Cre}Junb^{fl/fl}$ BM cells were equivalent to those derived from $Junb^{fl/fl}$ BM cells (Fig. 5b), indicating that cell-intrinsic

functions of JunB are not required for development of tTreg cells. In contrast, in spleen and lung, Foxp3$^+$ Treg cells derived from $Cd4^{Cre}Junb^{fl/fl}$ BM cells were significantly under-represented, compared with those derived from $Junb^{fl/fl}$ BM cells (Fig. 5b). Furthermore, the reduction of cells derived from $Cd4^{Cre}Junb^{fl/fl}$ BM was specific to the CD62L$^{lo}$ eTreg population, but not the CD62L$^{hi}$CD44$^{lo}$ cTreg population (Fig. 5c). We also confirmed impaired expression of ICOS and TIGIT in the remaining $Junb$-deficient CD62L$^{lo}$ eTreg cells (Fig. 5d, e), indicating that JunB regulates expression of ICOS and TIGIT in a cell-intrinsic manner.

The severe decrease of $Junb$-deficient eTreg cells in WT: $Cd4^{Cre}Junb^{fl/fl}$ mice suggests that JunB is important for eTreg homeostasis in a competitive setting. To further investigate this phenotype, we analyzed the effect of JunB deficiency on proliferation of Treg cells, using Ki67 staining. Proportions of Ki67-expressing eTreg cells derived from $Junb^{fl/fl}Cd4^{Cre}$ BM cells were severely decreased compared with those derived from $Junb^{fl/fl}$ BM cells (Fig. 5f). We also analyzed the effect of JunB deficiency on Treg cell death, using Annexin-V staining. Frequencies of Annexin-V-stained eTreg cells derived from $Junb^{fl/fl}Cd4^{Cre}$ BM cells, but not $Junb^{fl/fl}$ BM cells, were significantly increased (Fig. 5g). These data suggest that JunB intrinsically promotes proliferation and survival of eTreg cells, which is important for eTreg homeostasis in a competitive environment.

**JunB is required for a transcriptional program for eTregs**. To understand JunB-dependent transcriptional regulation in Treg cells, we performed RNA-sequence analysis of $Junb$-sufficient and $Junb$-deficient Treg cells, isolated from $Junb^{fl/fl}$ and $Cd4^{Cre}Junb^{fl/fl}$ mice, respectively. In total, 173 genes were downregulated and 137 genes were upregulated in $Junb$-deficient Treg cells compared with $Junb$-sufficient Treg cells ($p$ values < 0.05, log2 fold-changes > 0.5 or < 0.5) (Fig. 6a). Gene ontology analysis of genes differentially expressed in $Junb$-deficient and -sufficient Treg cells showed enrichment for genes associated with metabolic processes (Supplementary Fig. 8a). Furthermore, ingenuity pathway analysis (IPA) revealed that differentially expressed genes were associated most significantly with cell survival (Supplementary Fig. 8b). IPA also identified CD3 and CD28 signaling pathways as upstream regulators for genes affected by JunB deficiency (Supplementary Fig. 8c).

To assess the function of JunB in a transcriptional program for eTreg cells, we compared our data with previously identified eTreg-related genes (genes that ICOS$^+$ eTreg cells expressed more highly than did ICOS$^-$ eTreg cells)[18]. Among 762 eTreg-related genes, only 70, including *Icos*, *Tigit*, *Fgl2*, *Ctla4*, and *Maf*,

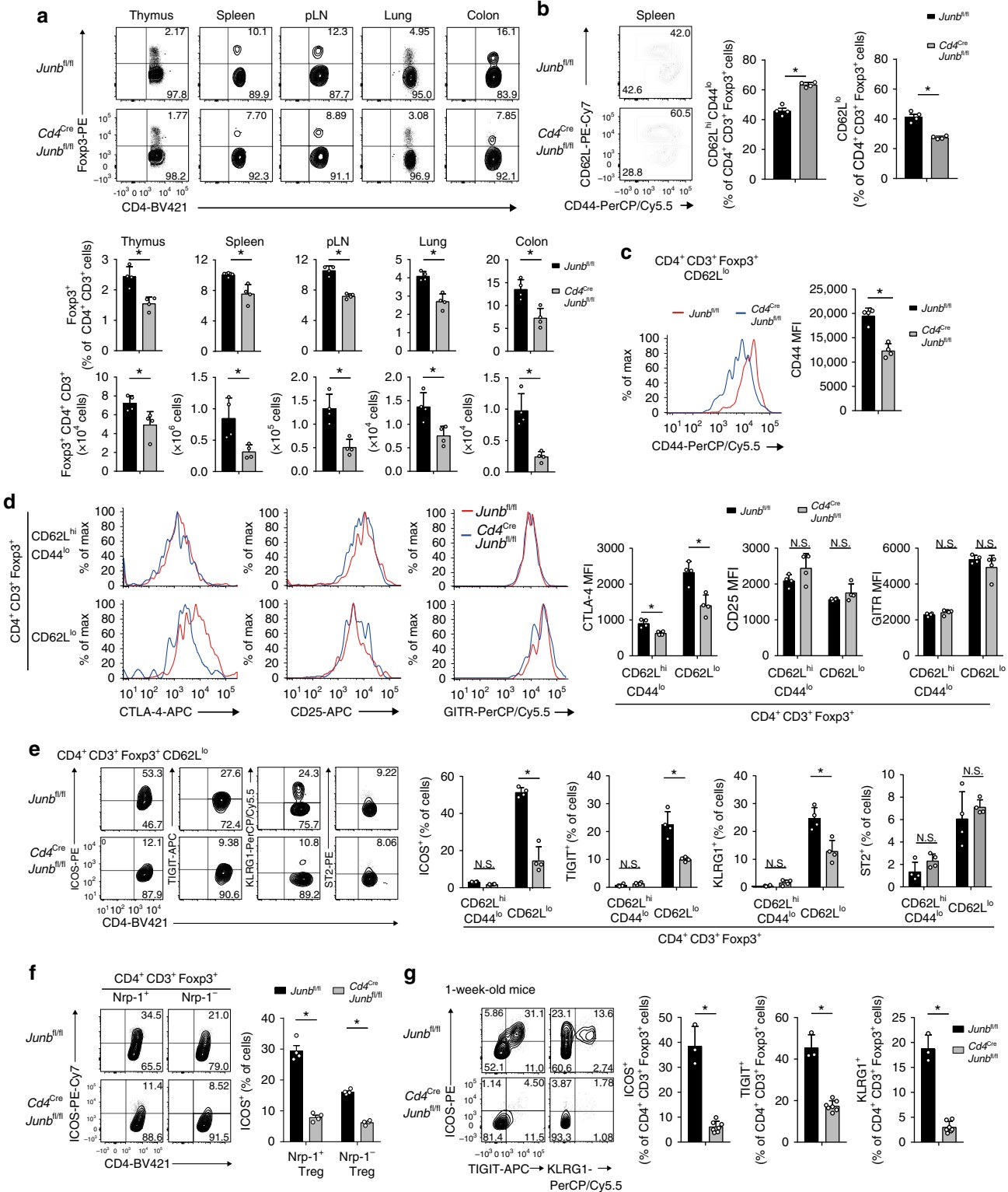

were downregulated in *Junb*-deficient Treg cells compared with *Junb*-sufficient Treg cells (Fig. 6a, b). We also compared our data with previously identified genes regulated by BATF in Treg cells[24]. Compared with JunB, many more eTreg-related genes (256 genes out of 762) were downregulated by BATF deficiency (Fig. 6b). Expression of only 34 eTreg-related genes, including *Icos* and *Fgl2*, was promoted by both JunB and BATF (Fig. 6b). In addition, expression of 36 eTreg-related genes, including *Ctla4*, *Tigit*, and *Maf* was upregulated by JunB, but not BATF, whereas

expression of 222 eTreg-related genes, including *Prdm1* and *Nt5e* was promoted by BATF, but not JunB (Fig. 6b). In addition, expression of *Il1rl1* (encoding ST2) and chemokine receptors *Ccr2*, *Ccr9*, and *Ccr10* was upregulated by BATF, but not JunB (Fig. 6b). These data indicate that JunB and BATF regulate shared and unique target genes in the eTreg transcriptional program.

To elucidate mechanisms underlying JunB-dependent transcriptional regulation, we conducted chromatin immunoprecipitation-sequencing (ChIP-Seq) analysis using

**Fig. 4** JunB regulates expression of eTreg-related molecules. **a** Flow cytometry analysis of CD4$^+$Foxp3$^+$ Treg cells in various tissues isolated from *Cd4*$^{Cre}$*Junb*$^{fl/fl}$ and *Junb*$^{fl/fl}$ mice. Graphs show percentages and numbers of CD4$^+$Foxp3$^+$ Treg cells. Error bars indicate s.d. ($n = 4$). *$P < 0.05$ (unpaired two-tailed Student's *t* test). **b**, **c** Flow cytometry analysis of CD44 and CD62L in CD4$^+$Foxp3$^+$ Treg cells isolated from spleens of *Cd4*$^{Cre}$*Junb*$^{fl/fl}$ and *Junb*$^{fl/fl}$ mice (8–12-week-old). Graphs show percentages of CD62$^{hi}$CD44$^{lo}$ cTreg and CD62$^{lo}$ eTreg cells **b**, and MFIs of CD44 **c**. Error bars indicate s.d. ($n = 4$). *$P < 0.05$ (unpaired two-tailed Student's *t* test). **d**, **e** Flow cytometry analysis of CTLA4, CD25, and GITR **d**, and ICOS, TIGIT, KLRG1, and ST2 **e** in CD62$^{hi}$CD44$^{lo}$ cTreg cells and CD62$^{lo}$ eTreg cells among CD4$^+$Foxp3$^+$ Treg cells isolated from spleens of *Cd4*$^{Cre}$*Junb*$^{fl/fl}$ and *Junb*$^{fl/fl}$ mice (8–12-week-old). Representative flow cytometry profiles show CD62$^{hi}$CD44$^{lo}$ cTreg cells and CD62$^{lo}$ eTreg cells **d**, and CD62$^{lo}$ eTreg cells **e**. Graphs show MFIs of CTLA4, CD25, and GITR **d**, and percentages of cells expressing indicated molecules **e**. Error bars indicate s.d. ($n = 4$). *$P < 0.05$; N.S., not significant (unpaired two-tailed Student's *t* test). **f**) Flow cytometry analysis of ICOS in Nrp1$^+$ and Nrp1$^-$ Treg cells isolated from spleens of *Cd4*$^{Cre}$*Junb*$^{fl/fl}$ and *Junb*$^{fl/fl}$ mice (8–12-week-old). The graph shows percentages of ICOS-expressing cells among Nrp1$^+$ and Nrp1$^-$ Treg cells. Error bars indicate s.d. ($n = 4$). *$P < 0.05$ (unpaired two-tailed Student's *t* test). **g** Flow cytometry analysis of ICOS, TIGIT, and KLRG1 in CD4$^+$Foxp3$^+$ Treg cells isolated from spleens of 1-week-old *Cd4*$^{Cre}$*Junb*$^{fl/fl}$ and *Junb*$^{fl/fl}$ mice. Graphs show percentages of cells expressing indicated molecules among CD4$^+$Foxp3$^+$ Treg cells. Error bars indicate s.d. ($n = 3$ for *Junb*$^{fl/fl}$, $n = 7$ for *Cd4*$^{Cre}$*Junb*$^{fl/fl}$). *$P < 0.05$ (unpaired two-tailed Student's *t* test). Data represent two independent experiments

antibodies against JunB, as well as against BATF and IRF4. We found that JunB-binding peaks overlapped significantly with BATF- and IRF4-binding peaks (Supplementary Fig. 9a). Among peaks identified around gene-encoding regions (within 100 kb of transcription start site), 35% of JunB peaks overlapped with BATF and IRF4 (Supplementary Fig. 9b). Furthermore, among peaks located around eTreg-related genes, higher numbers of JunB peaks (47%) overlapped with BATF and IRF4 (Supplementary Fig. 9c). As in previously reported data for Th17 and CD8$^+$ T cells[36–40], de novo motif analysis identified AICE motifs in overlapping ChIP-seq peaks for JunB, BATF, and IRF4 (Supplementary Fig. 9d).

We next investigated whether JunB is required for DNA-binding of IRF4 by ChIP-seq analysis for IRF4 in *Junb*-sufficient and *Junb*-deficient Treg cells. Global analysis of ChIP-seq peaks revealed that levels of IRF4 DNA-binding were significantly decreased in 28% of ChIP-seq peaks that overlapped for IRF4 and JunB (3190 out of 13,813 peaks, >2× changes). De novo motif analysis identified AP-1 and AICE motifs in overlapping ChIP-seq peaks for JunB and IRF4, in which levels of IRF4-binding were downregulated by JunB deficiency (supplementary Fig. 9e), implying the importance of JunB for DNA-binding of IRF4 at a subset of loci containing AICE motifs. We noted that at IRF4-binding sites located between *Icos* and *Ctla4*, both of which are upregulated by JunB and IRF4 in Treg cells, levels of some IRF4 peaks were reduced in *Junb*-deficient Treg cells (Fig. 6c). To confirm this, we performed ChIP for IRF4 followed by PCR analysis using primers to detect genomic regions in which significant decreases of DNA-binding of IRF4 by JunB deficiency were shown in ChIP-seq data. JunB deficiency considerably reduced IRF4 binding to sites located near *Icos*, *Ctla4*, *Gzmb*, and *Maf* (Fig. 6d and Supplementary Fig. 9f, g). In contrast, at IRF4-binding sites located near *Prdm1* and *Gata3*, DNA binding of IRF4 and BATF was not affected by JunB deficiency (Supplementary Fig. 9h, i). We also confirmed that JunB deficiency did not affect expression levels of IRF4 and BATF in eTreg cells (Supplementary Fig. 1e, f). These data suggest that JunB serves key functions in expression of a subset of IRF4-dependent, eTreg-related genes, probably by promoting DNA-binding of IRF4 at sites associated with those genes.

## Discussion

The TCR-dependent eTreg transcriptional program is indispensable for Treg cells to maintain immune homeostasis[6]. Upon TCR stimulation, IRF4 is induced and transcriptionally regulates a variety of eTreg-related genes[7]. In addition, an AP-1 transcription factor, BATF, which can interact with IRF4 to bind AICE motifs, is also required for eTreg differentiation[11,24,49]. However, it remains largely unclear how the IRF4-dependent eTreg program is regulated by AP-1 transcription factors. In this study, we show that the AP-1 transcription factor, JunB, as well as BATF and IRF4, are highly expressed in ICOS$^+$ eTreg cells. JunB is critical for accumulation and suppressive functions of eTreg cells, and Treg-specific ablation of JunB results in multi-organ autoimmunity. Unlike IRF4, which is absolutely necessary for CD62L$^{lo}$ eTreg generation, JunB regulates expression of a limited set of key effector molecules, such as ICOS and CTLA4, presumably by regulating DNA-binding activity of IRF4 in a locus-specific manner.

Our ChIP-seq data indicate that JunB colocalizes with BATF and IRF4 at loci of large numbers of eTreg-related genes containing AICE motifs. However, our RNA-seq data show that JunB is needed for expression of only a small subset of eTreg-related genes, such as *Icos*, *Klrg1*, *Tigit*, *Ctla4*, and *Gzmb*, in Treg cells. Interestingly, although JunB and BATF regulate some shared target genes, such as *Icos* and *Klrg1*, each regulates a unique set of genes (e.g., *Il1rl1* and *Prdm1* for BATF, and *Ctla4* and *Tigit* for JunB). JunB likely promotes accumulation of IRF4 at loci of selective eTreg-related genes, as loss of JunB severely decreases DNA-binding of IRF4 at its target sites located near *Icos*, *Ctla4*, *Gzmb*, and *Maf*. In addition, the necessity of JunB for IRF4-dependent gene expression is likely context-dependent. For example, expression of CTLA4 and TIGIT is decreased in eTreg cells derived from *Cd4*$^{Cre}$*Junb*$^{fl/fl}$ mice, but not from *Foxp3*$^{Cre}$*Junb*$^{fl/fl}$ mice, although expression of ICOS is decreased in eTreg cells derived from both types of mice. In contrast, expression of GATA3 and T-bet is diminished in eTreg cells derived from *Foxp3*$^{Cre}$*Junb*$^{fl/fl}$ mice, but not *Cd4*$^{Cre}$*Junb*$^{fl/fl}$ mice. These results suggest that the eTreg transcriptional program is differentially regulated in homeostatic and inflammatory environments. It remains unclear what determines the necessity of JunB for locus-specific DNA-binding of IRF4. Future studies are needed to identify differences in cis-regulatory elements or chromatin status associated with the necessity of JunB.

JunB is not required for development of tTreg cells, as accumulation of *Junb*-deficient Treg cells is comparable to accumulation of *Junb*-sufficient Treg cells in the thymus of BM chimera mice. However, JunB is required for accumulation of eTreg cells in a context-dependent manner. Even though a sizeable fraction of Treg cells differentiates into CD62L$^{lo}$ eTreg cells in *Foxp3*$^{Cre}$*Junb*$^{fl/fl}$ mice, accumulation of *Junb*-deficient eTreg cells, is severely decreased in the presence of *Junb*-sufficient Treg cells in WT:*Cd4*$^{Cre}$*Junb*$^{fl/fl}$ BM chimera mice. In WT:*Cd4*$^{Cre}$*Junb*$^{fl/fl}$ BM chimera mice, proliferation, and survival of *Junb*-deficient eTreg cells are significantly decreased, suggesting that JunB is essential for homeostasis of eTreg cells in a competitive setting under non-inflammatory conditions. Given that ICOS promotes survival of eTreg cells[14], deficient ICOS expression might partly explain impaired fitness in competitive environments for *Junb*-deficient eTreg cells.

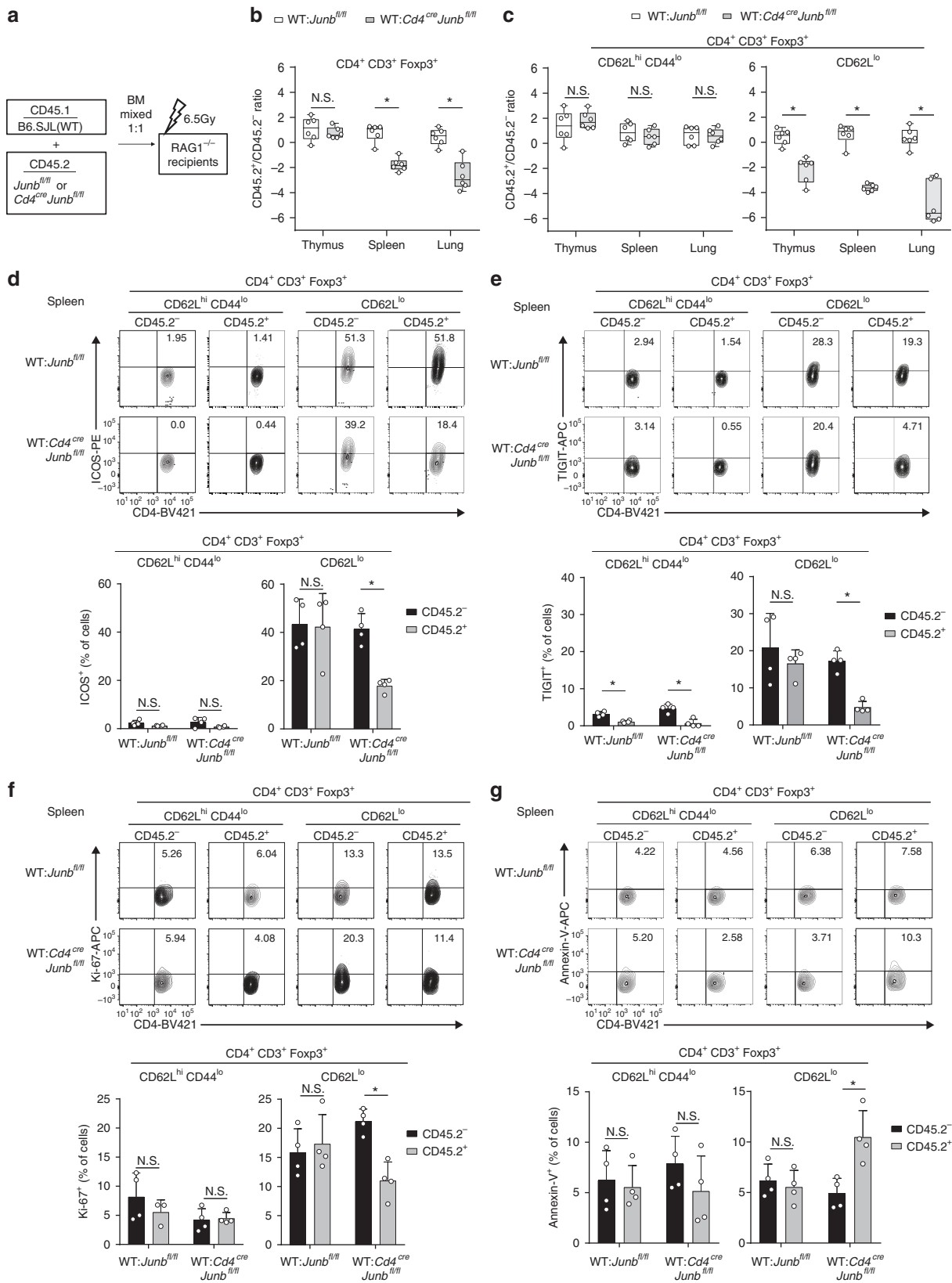

JunB is also important for accumulation of eTreg cells in peripheral tissues, particularly in the colon. As an adhesion molecule CD44 is involved in T-cell migration to inflammatory tissues[55], reduced expression of CD44 in *Junb*-deficient eTreg cells may affect migration of eTreg cells. Our in vitro and in vivo data indicate that JunB is also essential for TCR-dependent suppressive functions of Treg cells by promoting survival of Treg cells activated with strong and/or prolonged antigen signals. Defects in suppressive function in *Junb*-deficient Treg cells could be also attributed to decreased expression of CTLA4, which is

**Fig. 5** JunB is indispensable for eTreg homeostasis under competitive conditions. **a** Schematic of generation of bone marrow (BM) chimera mice. **b–g** Lethally irradiated 6–8-week-old sex-matched $Rag1^{-/-}$ mice were transferred with BM cells from $Cd4^{Cre}Junb^{fl/fl}$ (CD45.2) and $Junb^{fl/fl}$ (CD45.2) mice, in combination with equal numbers of WT BM cells from B6SJL (CD45.1), generating WT: $Cd4^{Cre}Junb^{fl/fl}$ and WT: $Junb^{fl/fl}$ BM chimera mice, respectively. Two to three months later, cells were isolated from thymuses, spleens, and lymph nodes, and ratios of CD45.2$^+$/CD45.2$^-$ in total CD4$^+$Foxp3$^+$ Treg cells **b**, CD62L$^{hi}$CD44$^{lo}$ cTreg cells and CD62L$^{lo}$ eTreg cells among CD4$^+$Foxp3$^+$ Treg cells **c** were determined by flow cytometry analysis. Error bars indicate s. d. ($n = 6$). *$P < 0.05$; N.S., not significant (unpaired two-tailed Student's $t$ test). **d–g** Flow cytometry analysis of ICOS **d**, TIGIT **e**, Ki67 **f**, and Annexin-V **g** in CD62L$^{hi}$CD44$^{lo}$ cTreg cells and CD62L$^{lo}$ eTreg cells among CD4$^+$Foxp3$^+$ Treg cells isolated from spleens of WT: $Cd4^{Cre}Junb^{fl/fl}$ and WT: $Junb^{fl/fl}$ BM chimera mice. Graphs show percentages of cells expressing indicated molecules or stained with Annexin-V. Error bars indicate s.d. ($n = 4$). *$P < 0.05$; N.S., not significant (unpaired two-tailed Student's $t$ test). Data represent two independent experiments

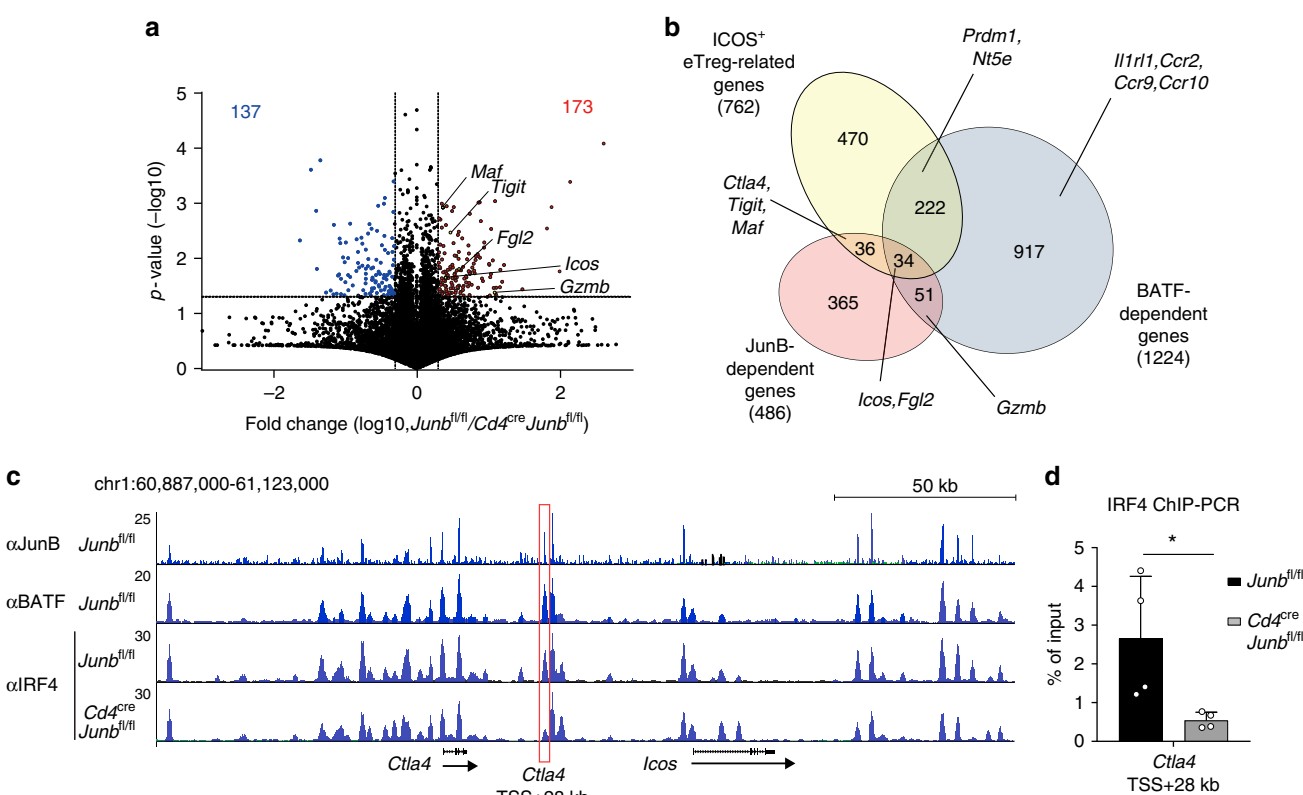

**Fig. 6** JunB diversifies targets for BATF and IRF4 in eTreg cells. **a–c** RNA-seq analysis of CD4$^+$CD25$^+$ Treg cells isolated from spleens of $Cd4^{Cre}Junb^{fl/fl}$ and $Junb^{fl/fl}$ mice. **a** Volcano plot comparing gene expression of Treg cells isolated from $Cd4^{Cre}Junb^{fl/fl}$ mice versus those from $Junb^{fl/fl}$ mice. **b** Venn diagram of genes upregulated by JunB (JunB-dependent genes) and BATF (BATF-dependent genes), and genes expressed in ICOS-expressing CD62L$^{lo}$ eTreg cells. **c** ChIP-seq analysis of CD4$^+$CD25$^+$ Treg cells isolated from spleens of $Cd4^{Cre}Junb^{fl/fl}$ and $Junb^{fl/fl}$ mice. Cells were stimulated with anti-CD3 and anti-CD28 antibodies in the presence of IL-2 for 60 h and used for ChIP-Seq analysis with antibodies against JunB, BATF, and IRF4. The genomic region containing $Ctla4$ and $Icos$ genes was shown. The red box indicates a region located 28 kb downstream of transcription start site of $Ctla4$ ($Ctla4$ TSS + 28 kb) in which the level of IRF4-binding was significantly decreased by JunB deficiency. Arrows indicate the direction of gene transcription. **d** ChIP-PCR analysis of CD4$^+$CD25$^+$ Treg cells isolated from spleens of $Cd4^{Cre}Junb^{fl/fl}$ and $Junb^{fl/fl}$ mice. Cells were stimulated with anti-CD3 and anti-CD28 antibodies in the presence of IL-2 for 72 h and used for ChIP with anti-IRF4 antibody. Immunoprecipitated DNA was analyzed by qPCR using primers to detect the IRF4-binding site located in the $Ctla4$ TSS + 28 kb region. Error bars indicate s.d. ($n = 4$). *$P < 0.05$ (unpaired two-tailed Student's $t$ test). Data represent two independent experiments

critical for Treg-dependent immune homeostasis[17,56]. In addition, our IPA analysis of genes affected by JunB deficiency suggested that TCR signaling and cell survival might be regulated by JunB. Thus, JunB-dependent modulation of the eTreg transcriptional program is critical for accumulation and suppressive functions of eTreg cells.

$Foxp3^{Cre}Junb^{fl/fl}$ mice induce multi-organ autoimmunity, particularly in lung and colon, whereas Treg-specific $Irf4$-deficient ($Foxp3^{Cre}Irf4^{fl/fl}$) mice develop autoimmune pathology in a broader array of tissues, including lung, colon, skin, liver, and pancreas[34]. Furthermore, although Th2 cells are preferentially

induced in $Foxp3^{Cre}Irf4^{fl/fl}$ mice[34], not only Th2 cells, but also Th1 and Th17 cells are aberrantly induced in $Foxp3^{Cre}Junb^{fl/fl}$ mice. In addition, $Foxp3^{Cre}Junb^{fl/fl}$ and $Foxp3^{Cre}Irf4^{fl/fl}$ mice have different sets of increased immunoglobulins. Although the difference in autoimmune pathology could be owing to distinct animal housing environments and different levels of leaky Cre expression in Tconv cells or CD8$^+$ T cells, it might also reflect functional differences between JunB and IRF4, discussed above.

eTreg cells express elevated levels of JunB, as well as BATF and IRF4, but regulatory mechanisms for their expression are somewhat different. In Treg cells, the CD3 signal is not sufficient to

induce JunB and BATF, but CD28 and IL-2 signals promote expression of JunB and BATF in an additive manner in the presence of CD3 signaling. In contrast, IRF4 expression is induced by CD3 signaling alone and is further augmented by CD28 and IL-2 signals, but there is no additive effect between CD28 and IL-2 signals. In addition, expression of BATF and IRF4 is transcriptionally upregulated, but JunB expression is regulated post-transcriptionally in activated Treg cells. The mechanism of post-transcriptional regulation of JunB expression may include JNK-phosphorylation-mediated stabilization of JunB protein[50]. Moreover, Treg cells express rather high levels of JunB, BATF, and IRF4 in lung compared with spleen, suggesting that tissue-specific signals modulate expression of JunB in peripheral tissues. These observations suggest that JunB, by cooperating with BATF and IRF4, integrates diverse signals to induce specific types of eTreg cells. Furthermore, heterogeneity in expression of these transcription factors can be generated during differentiation of eTreg cells, which might contribute to functional heterogeneity of eTreg cells.

In summary, we identify JunB as a key regulator of IRF4 activity in the eTreg transcriptional program. JunB serves non-redundant functions to facilitate expression of a subset of IRF4 target genes in BATF-dependent and -independent fashions. Thus, JunB could be a target for therapeutic manipulation of specific functions of eTreg cells in immune and autoimmune responses.

## Methods

**Mice.** $Junb^{fl/fl}$ mice were described previously[42]. $Foxp3^{Cre}$ (Foxp3$^{YFP-Cre}$; stock# 016959), $Cd4^{Cre}$ (stock# 017336), $Rag1^{-/-}$ (stock# 002216), and B6SJL (stock# 002014) mice were obtained from the Jackson Laboratory. All mice were maintained on a C57BL/6 background under specific pathogen-free conditions. Sex-matched, 6–12-week-old mice were used for experiments. All animal experimental protocols were approved by the Animal Care and Use Committee at Okinawa Institute of Science and Technology Graduate University.

**Antibodies.** For flow cytometry analysis and FACS, the following antibodies were used: anti-CD3 (17 A2, Biolegend, 1:400), anti-CD4 (GK1.5, Biolegend, 1:100 or 1:400), anti-CD8 (53–6.7, Biolegend, 1:400), anti-CD25 (PC61, Biolegend, 1:400), anti-CD44 (IM7, Biolegend, 1:400), anti-CD62L (MEL-14, Biolegend, 1:400), anti-CD19 (6D5, Biolegend, 1:400), anti-CD45R/B220 (RA3-6B2, Biolegend, 1:400), anti-Fas (SA367H8, Biolegend, 1:400), anti-GL-7 (GL7, Biolegend, 1:400), anti-CD138 (281–2, Biolegend, 1:400), anti-CTLA4 (UC10-4B9, Biolegend, 1:400), anti-GITR (DTA-1, Biolegend, 1:400), anti-ICOS (7E.17G9, Biolegend, 1:100 or 1:400), anti-TIGIT (1G9, Biolegend, 1:400), anti-KLRG1 (2F1/KLRG1, Biolegend, 1:400), anti-ST2 (DIH9, Biolegend, 1:200), anti-IL-17A (TC11-18H10.1, Biolegend, 1:500), anti-IFN-γ (XMG1.2, Biolegend, 1:100), anti-JunB (C-11, Santa Cruz Biotechnology, 1:200), anti-IRF4 (IRF4.3E4, Biolegend,1:100), anti-BATF (D7C5, Cell Signaling Technology, 1:400), anti-Foxp3 (150D, Biolegend, 1:100), anti-GATA3 (16E10A23, Biolegend,1:200), anti-RORγt (Q31-378, BD, 1:100), anti-T-bet (4B10, Biolegend, 1:100), anti-Helios (22F6, Biolegend, 1:100), anti-CD45.1 (A20, Biolegend, 1:400), anti-CD45.2 (104, Biolegend, 1:400), anti-Ki-67 (16A8, Biolegend, 1:400), anti-goat IgG (Poly4053, Biolegend, 1:100 or 1:400), and anti-rabbit IgG (Poly4064, Biolegend, 1:100 or 1:400). For ChIP analyses, anti-JunB (210, Santa Cruz, 2 μg per ChIP), anti-BATF (WW8, Santa Cruz, 2 μg per ChIP), and anti-IRF4 (M-17, Santa Cruz, 2 μg per ChIP) were used.

**Cell isolation.** Cells were isolated from spleens, lymph nodes, and thymuses by mashing them through cell strainers (BD; 352340). Cells were also isolated from lungs and colonic lamina propria using a Lung Dissociation kit (Miltenyi; 130-095⁻927) and a Lamina Propria Dissociation kit (Miltenyi; 130-397-410), respectively, according to the manufacturer's instructions.

**Flow cytometry.** For analysis of cell surface molecules, cells were stained in phosphate-buffered saline (PBS; Invitrogen; 21600) containing 2% fetal calf serum (Biosera; FB-I061) for 30 min on ice. For analysis of intracellular molecules, cells were stained with a Foxp3 Staining Buffer Set (eBioscience; 00-5253-00) according to the manufacturer's protocol. For analysis of intracellular cytokines, cells were re-stimulated with phorbol 12-myristate 13-acetate (Sigma; P8139; 50 ng mL$^{-1}$) and ionomycin (Sigma; I0634; 500 ng mL$^{-1}$) in the presence of brefeldin A (Biolegend; 420601; 5 μg mL$^{-1}$), and then stained with a Foxp3 Staining Buffer Set. Before antibody staining, cells were incubated with anti-Fc receptor-blocking antibody (anti-CD16/CD32; Biolegend; 101320) and NIR-Zombie (Biolegend; 423106). In

some experiments, dead cells were stained with Annexin-V (Biolegend, 1:400). The gating strategy for flow cytometry analysis and isotype control data are shown in Supplementary Fig. 10.

**Histological analysis.** Mouse lung, colon, liver, and skin tissues were fixed in 10% formalin, processed, and stained with hematoxylin and eosin. Inflammatory scores were determined by pathologists in the Biopathology Institute Co. Ltd., Japan.

**Mass cytometry analysis.** Cells isolated from mouse spleens were stained for viability in PBS containing 50 μM cisplatin (Fludigm; 201198). Cells were then incubated with anti-Fc receptor-blocking antibody (anti-CD16/CD32; Biolegend; 101320) for 10 min at room temperature, followed by staining with metal-conjugated antibodies for cell surface molecules (Maxpar Mouse Spleen/Lymph node Phenotyping Panel Kit; Fludigm; 201306) according to the manufacturer's protocol. Then, cells were fixed and permeabilized using a Foxp3 staining buffer set and subjected to staining with anti-Foxp3 antibody (Fludigm; FJK-16s-158Gd). Cells were suspended in Maxpar water supplemented with 10% EQ Four Element Calibration Beads (Fludigm; 201078) and analyzed on a Helios instrument (Fludigm) with Cytobank premium software.

**Enzyme-linked immunosorbent assay.** Serum was collected from 8–12-week-old male $Foxp3^{cre}Junb^{fl/fl}$ and control mice. Concentrations of IgE were determined with a Mouse IgE ELISA MAX Standard kit (Biolegend, 432401), and concentrations of IgG1, IgG2a, IgG2b, and IgA were determined as follows. Flat-bottom, 96-well plates (Greiner, 655061) were coated with the following capture antibodies overnight at 4ºC: anti-mouse IgA (C10-1, BD, 1:500), anti-mouse IgG2a (R11-89, BD, 1:500), anti-mouse IgG2b (R9-91, BD, 1:500), and anti-mouse IgG1 (A85-3, BD, 1:500). Plates were washed and blocked with 0.05% Tween20 in PBS (PBST) containing 5% bovine serum albumin (BSA, Wako, 018-15154) for 4 h at room temperature. The following standard immunoglobulins were used for quantification: IgG1 (S1-68.1, BD), IgG2a (C76-47, BD), IgG2b (C48-4, BD), and IgA (MOPC-320, BD). Samples were diluted with PBST containing 5% BSA (1:10,000 or 1:100,000 for IgA, 1:1,000,000 or 1:10,000,000 for IgG1, 1:120 or 1:1,200 for IgG2a, 1:100,000 or 1:1,000,000 for IgG2b, 1:100, 1:1000, 1:10,000, or 1:100,000 for IgE), added to plates, and incubated for 2 h at room temperature (for IgA and IgE) or overnight at 4ºC (for IgG1, IgG2a, and IgG2b). Plates were washed and incubated with the following biotin-conjugated secondary antibodies for 1 h at room temperature: anti-mouse IgA (C10-1, BD, 1:1000), anti-mouse IgG1 (A85-1, BD, 1:1000), anti-mouse IgG2a (R19-15, BD, 1:1000), and anti-mouse IgG2b (R12-3, BD, 1:1000). Plates were then incubated with streptavidin conjugated with horseradish peroxidase (BD, 554066, 1:1000) for 30 min at room temperature. Plates were then incubated with TMB (Sigma, T0440) for 10 min. The reaction was stopped by 2 N sulfuric acid, and absorbance at 450 nm and 570 nm was read using iMark plate reader (Bio-rad).

**RT-qPCR analysis.** Total RNA was isolated from FACS-sorted Treg cells using an RNeasy Plus Mini Kit (Qiagen; 74136). Complementary DNA was synthesized using a Revertra Ace qPCR Kit (Toyobo; FSQ-101) and subjected to qRT-PCR analysis with Faststart SYBR master mix (4673484, Roche) and a Thermal Cycler Dice Real Time system (Takara).

**RNA-sequencing analysis.** RNA-sequence libraries were prepared with NeoPrep (Illumina) using TruSeq Stranded mRNA NeoPrep Kit (Illumina). Libraries were then purified using Agencort AMPure XP (Beckman Coulter; A63880) at the ratio of 3:2 to remove adapter-dimer, then quantified with droplet digital PCR (Bio-Rad). Sequencing was performed on a HiSeq4000 (Illumina) with a HiSeq 3000/4000 SBS Kit (300 Cycles, Illumina; FC-410–1003) and a HiSeq 3000/4000 PE Cluster Kit (Illumina; PE-410-1001) to generate 150-nucleotide paired-end reads at a read depth of at least 20 million reads per sample.

**ChIP-seq and ChIP-PCR analyses.** ChIP samples were prepared using a SimpleChIP Plus Enzymatic Chromatin IP Kit (9005 S, Cell Signaling) as previously described. Treg cells (2–8 × 10$^5$ per ChIP-seq or 1–4 × 10$^5$ per ChIP-PCR) were activated with anti-CD3 antibody and anti-CD28 antibody in the presence of IL-2 for 60 h, cross-linked in culture medium containing 1% formaldehyde at room temperature for 10 min, and glycine solution was added to stop the reaction. After lysing cells, nuclei were isolated and treated with micrococcal nuclease (0.00313 μL mL$^{-1}$) for 20 min at 37℃, and the reaction was stopped by adding 0.05 M ethylene glycol-bis(β-aminoethyl ether)-N,N,N′,N′-tetraacetic acid (EGTA). Samples were then sonicated to disrupt nuclear membranes and centrifuged to collect supernatants containing chromatin. Chromatin solutions were incubated with 1 μg of antibodies overnight at 4℃ with rotation, and complexes of antibodies and chromatin were collected with Protein G magnetic beads (Veritas; DB10003). Beads were washed five times with low-salt wash solution and three times with high-salt wash solution buffer (incubated for 5 min for each washing) at 4℃. Chromatin was eluted, de-cross-linked following the manufacturer's instructions, purified by phenol/chloroform extraction, and used for ChIP-sequencing and

ChIP-PCR analyses. Primers used for ChIP-PCR were listed in Supplementary Table 1.

To prepare ChIP-seq libraries, immunoprecipitated DNA was blunt-ended and ligated with adaptors using a KAPA Hyper Prep Kit (KAPA Biosystems; KK8500). DNA was then cleaned up with an Agencort AMPure XP (Beckman Coulter; A63880) at a $1.8 \times$ DNA ratio, amplified by PCR, and purified using the AMPure XP at a $1.2 \times$ DNA ratio. Library DNA was size-selected using a 2% agarose gel cassette of Blue Pippin (Sage Science) for a target size range 150–300 bp, quantified with droplet digital PCR (Bio-Rad), and then sequenced on an Illumina HiSeq4000 to obtain 10 million uniquely aligned reads.

**In vitro Treg suppression assay.** Responder CD4$^+$ T cells were isolated from B6. SJL (CD45.1) mice and stained with CTV (Thermo Fisher Scientific; C34751). CTV-stained responder cells ($5 \times 10^4$ cells/well) were cultured alone or together with CD4$^+$CD25$^+$ Treg cells, isolated freshly or pre-activated by anti-CD3/CD28 antibodies plus IL-2 for 2 days, in the presence of anti-CD3/CD28-coated Dyna-beads (Invitrogen; 11456D; $1.25 \times 10^4$ beads/well). On day 3, cells were stained for CD45.1, CD45.2, and CD4 and analyzed by flow cytometry.

**In vivo Treg suppression assay.** Rag1$^{-/-}$ mice were intraperitoneally injected with CD4$^+$CD62L$^{hi}$CD44$^{lo}$CD25$^-$ naïve T cells ($4 \times 10^5$ cells per mouse) isolated from spleens and lymph nodes of wild-type C57BL/6 mice, together with or without CD4$^+$CD25$^{hi}$ Treg cells ($2 \times 10^5$ cells per mouse) isolated from spleens and lymph nodes of Cd4$^{Cre}$Junb$^{fl/fl}$ or Junb$^{fl/fl}$ mice. Mice were weighed to monitor disease progress.

**Mixed BM chimera mice.** Rag1$^{-/-}$ recipient mice were lethally irradiated by X-ray with a single dose of 6.5 Gy and 1 day later intravenously injected with a 1:1 mixture of BM cells from B6SJL (CD45.1) mice and Cd4$^{Cre}$Junb$^{fl/fl}$ or Junb$^{fl/fl}$ mice ($1 \times 10^7$ cells per mouse). Mice were given drinking water containing 1 mM Tri-methoprim (Sigma-Aldrich; 46984) and 5 mM Sulfamethoxazole (Sigma-Aldrich; 31737) for 1 week before and after irradiation. Mice were maintained for at least 2 months after transplantation and used for analysis.

**Statistical analysis.** Unpaired two-tailed Student's $t$ tests were performed using Prism software (GraphPad). $P$ values < 0.05 were considered statistically significant.

## Data availability

The RNA-seq and ChIP-seq data have been deposited in the Gene Expression Omnibus with the primary accession code GSE121295. CyTOF data were uploaded in Cytobank (this data will be available from the authors upon request). All other data supporting the findings of this study are available from the authors on request.

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

## Acknowledgements

We thank Dr. Steven D. Aird for editing the manuscript. We also thank our laboratory members and Dr. Taku Kureha for valuable discussions. This work was supported by KAKENHI grant (16K19164, 18K15201, 18K15200) and by OIST Graduate University.

## Author contributions

S.-i.K., D.S., and H.I. designed experiments, analyzed data, and wrote the manuscript. S.-i.K., D.S., T.-H.H, N.T., K.W., S.S., H.S., and M.M carried out experiments. N.A. and S.Y. conducted RNA-seq and ChIP-seq experiments.

## Additional information

**Competing interests:** The authors declare no competing interests.

