## [Peer Review File · Nature Communications]

Reviewers' comments:

Reviewer #1 (Helper T, Treg)(Remarks to the Author):

Koizumi and colleagues explore the role of the AP-1 transcription factor JunB in the development, function, and transcription program of effector regulatory T cells (eTregs). They find that JunB is necessary for eTreg survival and regulation of inflammation, driving transcription of a limited number of genes necessary for both, apparently in concert with IRF4. The data provided are generally clear, and the experiments well done.

The authors make the distinction between two populations of thymic-derived central Treg cells and effector Treg cells which they differentiate by the presence of CD62L. They present a model in which JunB mediates eTreg function, maintenance, and responsiveness to inflammatory signals, and, to a degree, eTreg development. This is in contrast to other key Treg transcription factors, such as IRF4, which are required for development of eTreg cells. Although discussed in the text, the idea that JunB regulates the functional responsiveness and maintenance of eTreg cells could be developed further, in that the mechanism by which JunB suppresses the generalized inflammatory responses observed in FoxP3 Cre JunB floxed mice is not clear. Several possibilities, discussed (or not) by the authors, are poor survival and/or proliferation due to ICOS deficiency, defective suppression secondary to reduced CTLA4, and reduced trafficking to proper effector sites due to reduced CD44. FoxP3 Cre JunB flox mice have enhanced numbers of myeloid cells and neutrophils in inflammatory lesions, increased numbers of Th cells, and elevated serum Igs, all unexplained. For example, more detailed analysis of the in vitro suppression and the adoptive transfer experiments (Fig 3 g, h) might help dissect these possibilities, such as molecular basis for the poor suppression, or altered survival, trafficking, or location of adoptively transferred cells.

It would be helpful to use isotype control antibodies in the flow plots. For example, in Figure 1, we assume JunB is not expressed in Tconv cells, as there is no difference in its expression in the CD4 Cre JunB flox vs. CD4 Cre strains. This assumption is based upon the notion of complete JunB deletion by the Cre, and the assumption there are no activated JunB bearing Tconv cells in non-immunized mice. Are these correct assumptions? What are the numbers of JunB eTregs? cTregs? Also, what are the Tconv cells here? How are they gated? Activated? Naïve?

In 2f, are B cell numbers reduced in FoxP3Cre JunB flox mice? If so, how does is this reconciled with increased amount of Igs (Fig. 2g)? It is not clear why the differential presence of inflammation in various organs in the absence of Treg JunB. A more careful analysis of cells present in end organs, in addition to analysis of the spleen, might be informative.

In Fig 3a, the finding of normal numbers of FoxP3+ Tregs in the lungs of the FoxP3Cre JunB flox mice is surprising, given the accumulation of eTregs in the lungs of naïve mice (Fig. 1). Are these cTregs? What do the Tconv cells look like in the lung? This might help explain the inflammation observed there. In Fig. 3b, why not examine the numbers of eTregs in peripheral tissues, for example the gut, where the inflammation is, in addition to the spleen?

It is not clear that the effect of JunB deletion is limited to peripheral Tregs. The data from the CD4 Cre JunB flox mice indicate deletion of thymic Tregs (Fig. 4a) although the BM chimera data (Fig. 5b) indicate the opposite. Also, the Y-axis in 4b is mislabeled. Shouldn't it be CD62L, instead of FoxP3?

In Fig. 6, the authors identify eTreg genes regulated by JunB, some of which are co-regulated by BATF or IRF4, with JunB promoting binding of IRF4 to some of its targets in eTreg cells. A functional analysis of the JunB-regulated genes is lacking, however, which might help illuminate the reason for

the inflammatory phenotype observed in the absence of JunB in FoxP3⁺ cells. For example, a gene ontology (GO) analysis would be helpful in helping understand the functional role of the genes regulated by JunB in eTregs. The authors have already demonstrated a functional significance of JunB in eTregs (Fig. 2). A GO analysis might help to elucidate the molecular pathways that underlie this functional importance.

Minor comments

Fig 1F: Should show the unstimulated control on the histograms, not just the bar graphs.

Fig 3C: Should label the x-axis as CD44 on the histogram.

Line 154: It should be "both"; line 155: should insert "compared to those".

Line 315: "eTre" should be "eTreg"

Reviewer #2 (Treg, TCR signaling)(Remarks to the Author):

This manuscript from Koizumi et al. examines the role of the transcription factor JunB in Treg development, homeostasis and function. The authors report some interesting findings regarding a specific role for JunB in the population of "effector" Treg (eTreg). For the most part, the data are novel and convincing, although there are some issues that the authors should address:

1. The sentence on lines 146-147 is rather vague. What are the authors trying to say here? Both here and in the Discussion, it may be helpful to have some reference to the fact that Jun family members are post-transcriptionally regulated, e.g. by the kinase JNK. This is especially important in light of the fact that they provide evidence for such post-transcriptional regulation later in the manuscript.
2. In Figure 2, the authors used FoxP3-YFP-Cre to knock out JunB. This Cre strain has been shown to be leaky under some circumstances (e.g. Franckaert et al. 2015 PubMed ID 25533288), so the authors should examine the relative specificity of deletion in Treg vs. other cell types. This becomes even more important in the light of the fact that in the CyTOF data myeloid cells seem to be most affected. Also, regarding the CyTOF, did the authors stain for JunB and what is its status in various compartments?
3. In Fig. 2, the authors show severe inflammation in the colon, raising the question of the status of the Peyer's patches; they should comment on this.
4. Fig. 2g shows elevated serum IgG in the KO animals. Is there spontaneous generation of germinal centers in these mice?
5. In Fig. 3f and 4f, the transcription factor staining is not convincing; nor is the TIGIT staining in Fig. 4e.
6. In Fig. 4g, the authors should state the basis of defining peripheral vs. thymic Treg. The text states that neuropilin-1 is one of the markers used. However, both neuropillin-1 and ICOS are known to be expressed in thymic as well as peripheral Treg.
7. In both the CD4-Cre and FoxP3-models, the authors show effects on CD4⁺ effector T cells. Is there also increased activation of CD8⁺ T cells?
8. With respect to the CyTOF and RNAseq data, are these data deposited in an accessible database?

This is important for fully evaluating the data and conclusions drawn from them.

Responses to reviewers' comments.

We greatly appreciate the constructive reviews provided by the reviewers. Considering all the reviewer's comments and suggestions, we have performed additional experiments, revised the manuscript text, and provided some new figures. Below, please find our point-by-point responses to questions and comments. We are confident that our paper has been considerably improved by the reviewers' critiques.

Reviewer #1:

Koizumi and colleagues explore the role of the AP-1 transcription factor JunB in the development, function, and transcription program of effector regulatory T cells (eTregs). They find that JunB is necessary for eTreg survival and regulation of inflammation, driving transcription of a limited number of genes necessary for both, apparently in concert with IRF4. The data provided are generally clear, and the experiments well done.

The authors make the distinction between two populations of thymic-derived central Treg cells and effector Treg cells which they differentiate by the presence of CD62L. They present a model in which JunB mediates eTreg function, maintenance, and responsiveness to inflammatory signals, and, to a degree, eTreg development. This is in contrast to other key Treg transcription factors, such as IRF4, which are required for development of eTreg cells. Although discussed in the text, the idea that JunB regulates the functional responsiveness and maintenance of eTreg cells could be developed further, in that the mechanism by which JunB suppresses the generalized inflammatory responses observed in FoxP3 Cre JunB floxed mice is not clear. Several possibilities, discussed (or not) by the authors, are poor survival and/or proliferation due to ICOS deficiency, defective suppression secondary to reduced CTLA4, and reduced trafficking to proper effector sites due to reduced CD44. FoxP3 Cre JunB

flox mice have enhanced numbers of myeloid cells and neutrophils in inflammatory lesions, increased numbers of Th cells, and elevated serum Igs, all unexplained. For example, more detailed analysis of the *in vitro* suppression and the adoptive transfer experiments (Fig 3 g, h) might help dissect these possibilities, such as molecular basis for the poor suppression, or altered survival, trafficking, or location of adoptively transferred cells.

We thank the reviewer for the positive and constructive comments. According to the reviewer's suggestion, we have extended the analysis of *Junb*-deficient Treg cells using *in vitro* suppression assays. We observed that anti-CD3 antibody stimulation strongly enhanced suppressive activity of

Junb-sufficient Treg cells, but not *Junb*-deficient Treg cells (Fig. 3f). In addition to this observation, our new data show that following stimulation with anti-CD3 antibody, frequency of Annexin-V⁺ cells was significantly increased in *Junb*-deficient Treg cells compared to controls (Supplementary Fig. 5a, lines 240-245). These results suggest that JunB is important for survival of Treg cells upon prolonged and/or strong TCR-stimulation, which promotes suppressive activity of Treg cells.

To further understand functions of JunB in Treg cells, we analyzed behavior of adoptively transferred *Junb*-deficient Treg cells. As with *Junb*-deficient Treg cells isolated from *Cd4^{Cre} Junb^{fl/fl}* mice (Fig. 3g), *Junb*-deficient Treg cells isolated from *Foxp3^{Cre} Junb^{fl/fl}* mice could not suppress weight loss induced by transfer of naïve CD4⁺ T cells in *Rag1*-deficient mice (Supplementary Fig. 5b, lines 246-250). Transferred *Junb*-deficient Treg cells accumulated, as did *Junb*-sufficient Treg cells, in the spleen, lymph nodes, and lungs, but their frequency in the colon was significantly lower than that of controls (Supplementary Fig. 5c, lines 250-254). These data are consistent with the observation in *Junb^{fl/fl} Foxp3^{Cre}* mice that accumulation of *Junb*-deficient Treg cells was specifically impaired in the gut (Fig. 3a).

In summary, our data suggest that JunB might be required for induction of full suppressive activity of Treg cells, probably by promoting survival of Treg cells activated with a strong and/or prolonged antigen signal. JunB is also likely important for migration and/or accumulation of Treg cells in the gut. Furthermore, under non-inflammatory conditions, JunB is involved in homeostasis of eTreg cells (Fig. 4). Thus, unlike IRF4, JunB is not essential for generation of eTreg cells, but it controls migration/accumulation and suppressive functions of eTreg cells probably by regulating IRF4 activity. We speculate that differences in autoimmune target organs, types of inflammatory cytokines and immunoglobulins observed in *Junb^{fl/fl} Foxp3^{Cre}* and reported *Irf4^{fl/fl} Foxp3^{Cre}* mice might be attributed to the function of IRF4-dependent, JunB-independent eTreg cells. For example, in *Junb^{fl/fl} Foxp3^{Cre}* mice, remaining JunB-independent eTreg cells may contribute to generate local cytokine environment favoring differentiation of not only Th2 cells but also Th1 and Th17 cells and to maintain immune homeostasis in certain tissues such as skin.

Although the molecular basis of JunB functions remains largely unclear, we have discussed possible physiological relevance of genes regulated by JunB in Treg cells. As already discussed, defective expression of ICOS might impair homeostasis of *Junb*-deficient eTreg cells. We have now also discussed possible impact of impaired expression of CD44 and CTLA4 expression in *Junb*-deficient eTreg cells, because CTLA4 is critical for Treg suppressive activity (Read *et al.*, *JEM*, 2000; Wing *et al.*, *Science*, 2008), and CD44 is involved in migration of T cells into inflammatory tissues (Degendele *et al.*, *Science*, 1997) (lines 427-433). Moreover, IPA analysis has revealed that TCR signal and cell survival might be regulated by JunB (details are discussed below), which is consistent with the observation that *Junb*-deficient Treg cells were extremely sensitive to cell death upon TCR stimulation (lines 433-435).

It would be helpful to use isotype control antibodies in the flow plots. For example, in Figure 1, we assume JunB is not expressed in Tconv cells, as there is no difference in its expression in the CD4 Cre JunB flox vs. CD4 Cre strains. This assumption is based upon the notion of complete JunB deletion by the Cre, and the assumption there are no activated JunB bearing Tconv cells in non-immunized mice. Are these correct assumptions? What are the numbers of JunB eTregs? cTregs? Also, what are the Tconv cells here? How are they gated? Activated? Naïve?

We have used isotype control antibodies for staining of ICOS, TIGIT, ST2, GATA3, ROR γ t, and T-bet (Supplementary Fig. 10, lines 515-516. We found that staining of wild-type naïve CD4⁺ T cells (which do not express JunB as discussed below) with the isotype control antibody for JunB gave significantly lower signals than staining with JunB antibody (Supplementary Fig. 10). Therefore, to avoid confusion, we used *Junb*-deficient CD4⁺ T cells as negative controls for FACS experiments of JunB in Figure 1.

Regarding JunB expression in Tconv cells, our previous immunoblot analysis showed that there was no detectable JunB expression in CD4⁺CD62L^{hi}CD44^{lo}CD25⁻ naïve CD4⁺ T cells, which represent majority of CD4⁺Foxp3⁻ Tconv cells (Hasan *et al.*, *Nat. Commun.*, 2017). To investigate whether CD4⁺Foxp3⁻ Tconv cells with an activated phenotype (CD62^{lo}CD44^{hi}) express JunB in the spleen of non-immunized mice, we compared JunB expression between naïve (CD62^{hi}CD44^{lo}) and activated (CD62^{lo}CD44^{hi}) Tconv cells. There was no difference in MFIs of JunB between Tconv cells bearing naïve and active phenotypes in the spleen (Supplementary Fig. 1a, line 124). These results indicate that Tconv cells in the spleen do not express detectable levels of JunB regardless of their activation status. In contrast, most CD4⁺Foxp3⁺ cells and CD4⁺Foxp3⁻ cells expressed high levels of JunB in the lungs of wild-type mice, but not in *Cd4^{cre}Junb^{fl/fl}* mice (Fig. 1b), suggesting that JunB is expressed in tissue-resident Tconv and Treg cells, and that Junb was efficiently deleted in the majority of CD4⁺ T cells in *Cd4^{cre}Junb^{fl/fl}* mice. We also reported that wild-type CD4⁺ T cells cultured under Th17-polarizing conditions, induce high levels of JunB expression, but CD4⁺ T cells isolated from *Cd4^{cre}Junb^{fl/fl}* mice do not express detectable levels of JunB, confirming effective knockout of JunB in these cells (Hasan *et al.*, *Nat. Commun.*, 2017).

To avoid confusion, we indicated definitions (gating strategies) of each cell population in all figures: Tconv (CD4⁺Foxp3⁻), Treg (CD4⁺Foxp3⁺), cTreg (CD4⁺Foxp3⁺CD62L^{hi}CD44^{lo}), eTreg (CD4⁺Foxp3⁺CD62L^{lo}). In addition, we have shown the gating strategy in Supplementary Fig. 10. Y-axis in bar graphs in Fig. 1 was indicated as “JunB MFI” instead of “MFI”.

Almost all cTreg cells did not express JunB, while more than 20% of eTreg cells expressed considerable levels of JunB (Fig. 1c).

In 2f, are B cell numbers reduced in FoxP3Cre JunB flox mice? If so, how does it reconcile with increased amount of Igs (Fig. 2g)? It is not clear why the differential presence of inflammation in various organs in the absence of Treg JunB. A more careful analysis of cells present in end organs, in addition to analysis of the spleen, might be informative.

To address behavior of B cells in *Foxp3^{Cre}Junb^{fl/fl}* mice, we analyzed frequencies and numbers of total B cells (B220⁺CD19⁺), germinal center B cells (B220⁺CD19⁺Fas⁺GL7⁺), and plasma cells (CD138⁺B220⁻) in the spleen, lymph nodes, and Peyer's patches. Frequencies of total B cells, germinal center B cells and plasma cells were significantly increased in cervical lymph nodes, but not in the spleen and Peyer's patches, in *Foxp3^{Cre}Junb^{fl/fl}* mice compared to controls (Supplementary Fig. 3a-c, lines 179-183). Thus, our data indicate that enhanced production of immunoglobulin is accompanied by increased numbers of germinal center B cells in the peripheral lymph nodes in diseased *Foxp3^{Cre}Junb^{fl/fl}* mice.

We agree that the cellular and/or molecular basis for organ-specific inflammation in *Foxp3^{Cre}Junb^{fl/fl}* mice remains unclear. According to the reviewer's suggestion, we analyzed abundance of Foxp3⁺ Treg cells and CD62L^{lo} eTreg cells in lung, colon, and skin of *Foxp3^{Cre}Junb^{fl/fl}* mice. CD4⁺Foxp3⁺ Treg cells were normally abundant in the spleen, lung, liver and skin, but they are severely decreased in the colon (Fig. 3a and Supplementary Fig. 4a, lines 203-210). In the colon, almost all Treg cells exhibited the CD62L^{lo} eTreg phenotype in both *Foxp3^{Cre}Junb^{fl/fl}* and *Foxp3^{Cre}Junb^{+/+}* mice (Supplementary Fig. 4b, lines 213-215). However, due to the decrease of total Treg cells, numbers of eTreg cells were severely decreased in colons of *Foxp3^{Cre}Junb^{fl/fl}* mice (Supplementary Fig. 4b, lines 213-215). In contrast, in lungs of *Foxp3^{Cre}Junb^{fl/fl}* mice, in which severe inflammation was observed, frequency of eTreg cells was much higher than that of control mice (Supplementary Fig. 4b, lines 213-215). In the skin in *Foxp3^{Cre}Junb^{fl/fl}* mice, in which inflammation was not observed, frequency of eTreg cells was comparable to that of control mice (Supplementary Fig. 4b, lines 213-215). In summary, CD62L⁻ eTreg cells were normally generated in many of lymphoid and non-lymphoid organs, except for the gut, in *Foxp3^{Cre}Junb^{fl/fl}* mice. This also supports our claim that JunB is necessary for suppressive functions of eTreg cells, but not for generation of eTreg cells. However, the reason why inflammation is not induced in certain tissues, such as skins, is currently unknown. We can only speculate that, depending on the tissue environment, eTreg cells rely on different mechanisms/molecules to maintain immune homeostasis.

In Fig 3a, the finding of normal numbers of FoxP3+ Tregs in the lungs of the FoxP3Cre JunB flox mice is surprising, given the accumulation of eTregs in the lungs of naïve mice (Fig. 1). Are these cTregs? What do the Tconv cells look like in the lung? This might help explain the inflammation observed there. In Fig. 3b, why not examine the numbers of eTregs in peripheral tissues, for example

the gut, where the inflammation is, in addition to the spleen?

As mentioned above, we added new data showing the frequency of eTreg cells in lung, colon and skin in *Foxp3^{Cre}Junb^{fl/fl}* mice. CD62L⁻ eTreg cells, but not CD62L⁺ cTreg cells, in the lung were more abundant in *Foxp3^{Cre}Junb^{fl/fl}* mice than in *Foxp3^{Cre}Junb^{+/+}* mice (Supplementary Fig. 4b, lines 213-215). In the lung of *Foxp3^{Cre}Junb^{fl/fl}* mice, Tconv cells with an activated phenotype (CD62^{lo}CD44^{hi}) were also significantly increased (Supplementary Fig. 2d, lines 176-179).

It is not clear that the effect of JunB deletion is limited to peripheral Tregs. The data from the CD4 Cre JunB flox mice indicate deletion of thymic Tregs (Fig. 4a) although the BM chimera data (Fig. 5b) indicate the opposite. Also, the Y-axis in 4b is mislabeled. Shouldn't it be CD62L, instead of FoxP3?

We analyzed expression of JunB in several immune cell populations by CyTOF. JunB was expressed mainly in granulocytes, and the expression levels of JunB in these cells were comparable between *Foxp3^{Cre}Junb^{+/+}* and *Foxp3^{Cre}Junb^{fl/fl}* mice, confirming that increased granulocytes in *Foxp3^{Cre}Junb^{fl/fl}* mice was not due to non-specific deletion of *Junb* in these cells (New Fig. 2f, lines 183-186). In the CyTOF analysis, we realized that there was a consistent increase of granulocytes in all mice analyzed, but that increases of macrophages/monocytes were limited to a subset of mice analyzed (confirmed in two independent experiments); thus, we have changed the text and Figure 2f (lines 183-186). We are sorry for the confusion.

To further investigate effects of JunB deficiency on Treg development, we analyzed the frequency of Treg cells in the thymus of *Cd4^{Cre}Junb^{fl/fl}* neonates (1-week-old mice). There was no difference in frequency of Treg cells in the thymus between *Cd4^{Cre}Junb^{fl/fl}* and *Junb^{fl/fl}* neonates (Supplementary Fig. 6c, lines 288-289). Taken together with the observation in the bone marrow chimera experiments (Fig. 5), these new data suggest that JunB is not critical for Treg development in the thymus. A slight decrease of Treg cells in the thymus of adult *Cd4^{Cre}Junb^{fl/fl}* mice suggests that the Treg-extrinsic role of JunB (probably in thymocytes or T cells) may affect abundance of *Junb*-deficient thymic Treg cells.

We have corrected the Y-axis in Fig. 4b.

In Fig. 6, the authors identify eTreg genes regulated by JunB, some of which are co-regulated by BATF or IRF4, with JunB promoting binding of IRF4 to some of its targets in eTreg cells. A functional analysis of the JunB-regulated genes is lacking, however, which might help illuminate the reason for the inflammatory phenotype observed in the absence of JunB in FoxP3⁺ cells. For example, a gene ontology (GO) analysis would be helpful in helping understand the functional role

of the genes regulated by JunB in eTregs. The authors have already demonstrated a functional significance of JunB in eTregs (Fig. 2). A GO analysis might help to elucidate the molecular pathways that underlie this functional importance.

We conducted GO analysis of genes differentially expressed in *Junb*-deficient and -sufficient Treg cells and found enrichment for genes associated with metabolic processes and cell cycle (Supplementary Fig. 8a, lines 331-333). We also performed ingenuity pathway analysis (IPA) and found that the differentially expressed genes in *Junb*-deficient and -sufficient Treg cells were associated most significantly with cell survival (Supplementary Fig. 8b, lines 333-335). In IPA, we also identified CD3 and CD28 signaling pathways as upstream regulators for genes affected by JunB deficiency (Supplementary Fig. 8c, lines 335-336).

Minor comments

Fig 1F: Should show the unstimulated control on the histograms, not just the bar graphs.

Done.

Fig 3C: Should label the x-axis as CD44 on the histogram.

Done.

Line 154: It should be “both”; line 155: should insert “compared to those”.

Done.

Line 315: “eTre” should be “eTreg”

Sorry for the typo. We have corrected it.

Reviewer #2 (Treg, TCR signaling)(Remarks to the Author):

This manuscript from Koizumi et al. examines the role of the transcription factor JunB in Treg development, homeostasis and function. The authors report some interesting findings regarding a specific role for JunB in the population of “effector” Treg (eTreg). For the most part, the data are novel and convincing, although there are some issues that the authors should address:

We thank the reviewer for the valuable comments.

1. The sentence on lines 146-147 is rather vague. What are the authors trying to say here? Both here

and in the Discussion, it may be helpful to have some reference to the fact that Jun family members are post-transcriptionally regulated, e.g. by the kinase JNK. This is especially important in light of the fact that they provide evidence for such post-transcriptional regulation later in the manuscript.

We appreciate this suggestion. We have changed the sentence around lines 146-147 in the previous manuscript (lines 146-150). Here we mentioned the reason why we examined JunB expression in TCR-stimulated Treg cells.

We also added the reference for post-transcriptional regulation of JunB expression (lines 139-140 and lines 455-457).

2. In Figure 2, the authors used Foxp3-YFP-Cre to knock out JunB. This Cre strain has been shown to be leaky under some circumstances (e.g. Franckaert et al. 2015 PubMed ID 25533288), so the authors should examine the relative specificity of deletion in Treg vs. other cell types. This becomes even more important in the light of the fact that in the CyTOF data myeloid cells seem to be most affected. Also, regarding the CyTOF, did the authors stain for JunB and what is its status in various compartments?

According to the reviewer's suggestion, we analyzed expression of JunB in major immune cell populations by CyTOF. JunB was highly expressed in granulocytes, and expression levels of JunB in these cells were not affected by Foxp3-driven Cre. (New Fig. 2f, lines 183-186). In these experiments, we observed consistent increases of granulocytes, but not macrophages/monocytes (confirmed in two independent experiments). Therefore, we have changed the text and Figure 2f. We are sorry for the confusion.

3. In Fig. 2, the authors show severe inflammation in the colon, raising the question of the status of the Peyer's patches; they should comment on this.

Our FACS analysis showed that frequencies of total B cells (B220⁺CD19⁺), germinal center B cells (B220⁺CD19⁺Fas⁺GL7⁺), and plasma cells (CD138⁺B220⁻) were comparable between *Foxp3*^{Cre} *Junb*^{fl/fl} mice and controls in the Peyer's patches (Supplementary Fig. 3a-c, lines 179-183).

4. Fig. 2g shows elevated serum IgG in the KO animals. Is there spontaneous generation of germinal centers in these mice?

We observed an increase of germinal center B cells (B220⁺CD19⁺Fas⁺GL7⁺) in the lymph nodes, suggesting spontaneous generation of germinal centers (Supplementary Fig. 3a-c, lines 179-183).

5. In Fig. 3f and 4f, the transcription factor staining is not convincing; nor is the TIGIT staining in Fig. 4e.

We tried to optimize the staining conditions, but could not get better signal-to-noise ratios, probably due to low levels of expression of those molecules, as reported before (Hayatsu *et al.*, *Immunity*, 2017). To make it more convincing, we added the data with isotype control isotype antibodies (Supplementary Fig. 10, lines 515-516). However, as expression of T-bet, GATA3 and ROR γ t (old Fig. 3f and Fig. 4f) was only modestly increased in Treg cells of *Junb*^{fl/fl}*Foxp3*^{Cre} mice and *Junb*^{fl/fl}*Cd4*^{Cre} mice, and we are not sure about the physiological significance of these effects, we moved the data to supplementary information.

6. In Fig. 4g, the authors should state the basis of defining peripheral vs. thymic Treg. The text states that neuropilin-1 is one of the markers used. However, both neuropillin-1 and ICOS are known to be expressed in thymic as well as peripheral Treg.

In addition to neuropillin-1, we used Helios, another marker to distinguish thymus-derived Treg cells and Treg cells induced in the periphery. JunB deficiency decreased eTreg cells in both Helios⁺ and Helios⁻ Treg cells (Supplementary Fig. 6b, lines 281-285), supporting our conclusion that JunB is required for ICOS induction in both thymus-derived Treg cells and Treg cells induced in the periphery.

To provide accurate information, in Fig. 4g, we provided the definition (gating strategy) of each cell population: NRPI⁺ Treg and NRPI⁻ Treg cells instead of tTreg and pTreg, respectively.

7. In both the CD4-Cre and FoxP3-models, the authors show effects on CD4⁺ effector T cells. Is there also increased activation of CD8⁺ T cells?

Our new data show a significant increase of CD8⁺ T cells with an activated phenotype (CD62L^{lo}CD44^{hi}) in spleen and lung of *Foxp3*^{Cre}*Junb*^{fl/fl} mice (Supplementary Fig. 3d, lines 193-196). Expression of inflammatory cytokines (IFN- γ and IL-17A) in CD8⁺ T cells was also upregulated in spleen and lung of *Foxp3*^{Cre}*Junb*^{fl/fl} mice, compared to control mice (Supplementary Fig. 3e, lines 193-196).

8. With respect to the CyTOF and RNAseq data, are these data deposited in an accessible database? This is important for fully evaluating the data and conclusions drawn from them.

Done (lines 618-620).

Reviewers' comments:

Reviewer #1 (Remarks to the Author):

(No specific comments).

Reviewer #2 (Remarks to the Author):

The authors have done a reasonably good job responding to my original critiques.

However, one major issue that has not been addressed adequately is the question of potential leakiness of the FoxP3-YFP-Cre mouse line. This is critical for proper interpretation of their findings.

Thus, the authors seem to have partly answered the question by showing that granulocytes in the spleen do not lose JunB. However, in other studies the leak is significantly higher in the conventional CD4 and CD8 compartment. Although the authors state that splenic CD4 and CD8 normally express little-to-no JunB, they also show (Fig. 1b) that conventional CD4 cells in the lung significantly upregulate JunB. The authors should therefore examine lung and gut for JunB levels on conventional CD4 and CD8 T cells to show that JunB downregulation is Treg-specific). For this experiment, it would be advisable to have splenic conventional CD4 T cells as a negative control.

Responses to reviewers' comments.

We appreciate the helpful reviewers' comment to improve our manuscript. Below please find our point-by-point response to the comment. All the changes in the manuscript are marked in red.

Reviewer #2 (Remarks to the Author):

The authors have done a reasonably good job responding to my original critiques.

However, one major issue that has not been addressed adequately is the question of potential leakiness of the Foxp3-YFP-Cre mouse line. This is critical for proper interpretation of their findings.

Thus, the authors seem to have partly answered the question by showing that granulocytes in the spleen do not lose JunB. However, in other studies the leak is significantly higher in the conventional CD4 and CD8 compartment. Although the authors state that splenic CD4 and CD8 normally express little-to-no JunB, they also show (Fig. 1b) that conventional CD4 cells in the lung significantly upregulate JunB. The authors should therefore examine lung and gut for JunB levels on conventional CD4 and CD8 T cells to show that JunB downregulation is Treg-specific). For this experiment, it would be advisable to have splenic conventional CD4 T cells as a negative control.

We are sorry that we did not address the issue in the previous manuscript. We have now analyzed JunB expression in Foxp3⁺CD4⁺ Treg cells, Foxp3⁻CD4⁺ Tconv cells and CD8⁺ T cells in spleen, gut and lung of Foxp3^{Cre}Junb^{fl/fl} mice (Supplementary Fig. 2a, lines 167-171). We observed substantial numbers of CD4⁺Foxp3⁻ Tconv cells and CD8⁺ T cells expressed JunB in the lung and colon of Foxp3^{Cre}Junb^{fl/fl} mice, albeit reduced relative to controls, probably due to leaky expression of Cre in these cells (Franckaert et al. 2015). As we cannot rule out the possibility that levels of leaky expression of Cre in Tconv cells and CD8⁺ T cells can affect autoimmune pathology, we mentioned it also in the discussion (lines 450-451).

REVIEWERS' COMMENTS:

Reviewer #2 (Remarks to the Author):

The authors have responded reasonably well to my one remaining critique. Having this additional clarification is an important addition to the manuscript.

Response to reviewers' comments.

REVIEWERS' COMMENTS:

Reviewer #2 (Remarks to the Author):

The authors have responded reasonably well to my one remaining critique. Having this additional clarification is an important addition to the manuscript.

Thank you very much for your time. We agree that the new data is important for the manuscript.